# Characterizing biospheric carbon balance using $CO_2$ observations from the OCO-2 satellite

Scot M. Miller[1,2], Anna M. Michalak[1], Vineet Yadav[3], and Jovan M. Tadić[4]

[1]Department of Global Ecology, Carnegie Institution for Science, Stanford, CA, USA
[2]Now at the Department of Environmental Health and Engineering, Johns Hopkins University, Baltimore, MD, USA
[3]Jet Propulsion Laboratory, California Institute of Technology, Pasadena, CA, USA
[4]Lawrence Berkeley National Laboratory, Berkeley, CA, USA

*Correspondence to:* Scot M. Miller (scot.m.miller@gmail.com)

**Abstract.**

NASA's Orbiting Carbon Observatory–2 (OCO-2) satellite launched in summer of 2014. Its observations could allow scientists to constrain $CO_2$ fluxes across regions or continents that were previously difficult to monitor. This study explores an initial step toward that goal; we evaluate the extent to which current OCO-2 observations can detect patterns in biospheric $CO_2$ fluxes and constrain monthly $CO_2$ budgets. Our goal is to guide top-down, inverse modeling studies and identify areas for future improvement. We find that uncertainties and biases in the individual OCO-2 observations are comparable to the atmospheric signal from biospheric fluxes, particularly during northern hemisphere winter when biospheric fluxes are small. A series of top-down experiments indicate how these errors affect our ability to constrain monthly biospheric $CO_2$ budgets. We are able to constrain budgets for between two and four global regions using OCO-2 observations, depending on the month, and we can constrain $CO_2$ budgets at the regional level (i.e., smaller than seven global biomes) in only a handful of cases (16% of all regions and months). The potential of the OCO-2 observations, however, is greater than these results might imply. A set of synthetic data experiments suggests that retrieval errors have a salient effect. Advances in retrieval algorithms and to a lesser extent atmospheric transport modeling will improve the results. In the interim, top-down studies that use current satellite observations are best-equipped to constrain the biospheric carbon balance across only continental or hemispheric regions.

# 1 Introduction

The OCO-2 satellite launched on July 2nd, 2014 and is NASA's first mission dedicated to observing $CO_2$ from space. The satellite measures the absorption of reflected sunlight within $CO_2$ and molecular oxygen ($O_2$) bands at near infrared wavelengths. These measurements are analyzed with remote sensing retrieval algorithms to yield spatially-resolved estimates of the column-averaged $CO_2$ dry air mole fraction, $XCO_2$. The satellite flies in a sun-synchronous orbit an average of 705 km above the Earth's surface, passing each location at approximately 13:30 local time, and it collects roughly $5 \times 10^5$ to $1 \times 10^6$ observations or soundings per calendar day (e.g., Crisp et al., 2004; Eldering et al., 2012; Crisp et al., 2017).

OCO-2 may provide an ideal opportunity for estimating surface $CO_2$ fluxes. OCO-2 observes in the near-infrared, and its observations therefore have sensitivity throughout the entire troposphere with highest sensitivity near the surface (e.g., Eldering et al., 2017a). This feature contrasts with several existing satellites that observe in the thermal infrared and have little sensitivity to near-surface $CO_2$ variations. OCO-2 also has a smaller footprint and improved spatial coverage relative to the Greenhouse Gas Observing Satellite (GOSAT). Each OCO-2 observation has a footprint ∼2.25km in width, and the satellite can collect eight observations across a single swath (Eldering et al., 2017a). Each GOSAT observation, by contrast, has a footprint ∼10km in width, and the satellite collects a single sounding every 250km (Yokota et al., 2009). As a result of these differences, GOSAT provides approximately 1000 high quality observations per day while OCO-2 provides approximately 65000 (e.g., Eldering et al., 2017b).

Prior to the OCO-2 satellite launch, several studies commented on the possibilities of using $XCO_2$ observations for estimating $CO_2$ fluxes at the Earth's surface. For example, Chevallier et al. (2007) and Baker et al. (2010) explain that OCO-2 observations could reduce flux uncertainties by ∼20–65% at the model grid scale (2.5° by 3.75° and 2° by 5° latitude-longitude, respectively) and at weekly time scales. Both studies, however, caution that biases or spatially and temporally correlated errors would cut this uncertainty reduction in half. Chevallier et al. (2007) further explain that biases of a few tenths of a part per million in $XCO_2$ could bias estimated subcontinental flux totals by several tenths of a gigaton.

Since launch, a handful of existing studies apply the satellite observations to estimate fluxes for specific problems. Most examine flux anomalies (e.g., El Niño) or anomalously large sources (e.g., power plants). For example, Chatterjee et al. (2017), Patra et al. (2017), Heymann et al. (2017), and Liu et al. (2017) estimate flux anomalies during the most recent El Niño, and Nassar et al. (2017) estimate emissions from several large power plants.

A number of studies use GOSAT observations to estimate surface fluxes, and these studies report numerous successes and challenges that could apply to OCO-2 (e.g., Takagi et al., 2011; Basu et al., 2013; Guerlet et al., 2013; Maksyutov et al., 2013; Parazoo et al., 2013; Saeki et al., 2013; Basu et al., 2014; Deng et al., 2014; Houweling et al., 2015). GOSAT observations provide new insight into fluxes in regions that are poorly sampled by in situ observations. These studies also identify a number of common challenges. For example, observations are too sparse to reliably estimate fluxes in regions with frequent cloud cover (e.g., Parazoo et al., 2013). Furthermore, continental $CO_2$ budgets estimated using GOSAT observations are not consistent with in situ observations in some regions; these differences may indicate spatially and temporally correlated errors in GOSAT

observations at their current stage of development (e.g., Houweling et al., 2015). These challenges may also be a concern for OCO-2.

This study evaluates the opportunities and challenges of using current OCO-2 observations to estimate biospheric $CO_2$ fluxes. A primary goal of this work is to guide top down, inverse modeling studies on the information content of currently-available observations. By contrast, satellite capabilities for $CO_2$ monitoring will likely change quickly over the next ten years – both due to improvements in satellite retrieval algorithms and the launch of new satellites (Sect. 4). This guidance will therefore undoubtedly change and evolve in the future.

We evaluate current OCO-2 observations using several approaches. We first compare model and observation errors against the atmospheric signal from biospheric fluxes. This initial comparison provides context and intuition for the signal-to-noise ratio. Ultimately, atmospheric inversions use more complex patterns in the observations to estimate surface fluxes. We therefore construct a series of top-down simulations using OCO-2 observations to understand what these errors mean for estimating $CO_2$ fluxes, and we thereby evaluate the number of global regions for which we can independently constrain $CO_2$ budgets. Lastly, we construct a series of synthetic data simulations to diagnose the real data results. The first synthetic simulations do not include any errors – to evaluate the inherent strengths of the observations. In subsequent synthetic simulations, we include simulated modeling and/or retrieval errors, and evaluate how these errors affect the $CO_2$ flux constraint.

## 2   Methods

### 2.1   OCO-2 data

This study utilizes $XCO_2$ observations from the OCO-2 satellite beginning with the first reported data (6 Sept. 2014) through the end of 2015. We use the level 2 lite product, version B7.1.01; the lite product only includes good quality retrievals, unlike the full OCO-2 level 2 product. We only include nadir and target mode retrievals in the analysis and exclude glint mode retrievals. Recent work indicates biases in the glint retrievals relative to nadir retrievals (e.g., Wunch et al., 2017). The SI describes model selection results with glint mode retrievals included, and the results are similar to those in the main manuscript without glint mode data.

### 2.2   Simulated model and retrieval errors

The simulated model and retrieval errors provide an intuitive feel for the observations and are used as inputs in the top-down experiments later on in the study. This section describes these simulated errors – both simulated atmospheric transport and retrieval errors (Fig. 1). The SI describes these errors in greater detail.

We use estimated $CO_2$ transport errors from Liu et al. (2011) and Miller et al. (2015) (Fig. 1a–b). The authors of those studies run an ensemble of global meteorology simulations. $CO_2$ is included as a passive tracer in the model, and all simulations use the same $CO_2$ flux estimate but have different meteorology. The authors estimate $CO_2$ transport errors by examining the range of $CO_2$ mixing ratios in the ensemble of simulations. We choose one realization at random and subtract the mean of the ensemble

from this realization to produce a set of residuals. These residuals are used as the estimated transport errors in this study. The estimated errors are therefore a realization of plausible transport errors. As a result, the specific errors used here have the same statistical properties as the other members of the ensemble but have different values in specific locations or at specific times. For example, the transport errors have a negative value across much of the Arctic in Fig. 1a. Other ensemble members might

have a slight positive bias in that region but will nevertheless have similar statistical properties as the realization in Fig. 1a.

In addition to these transport errors, we simulate retrieval errors. We use two different approaches to estimate these errors and report results using both approaches. The true retrieval errors are unknown and any effort to estimate these errors will be uncertain; the two approaches used here provide two contrasting, plausible representations of these errors.

We generate the first set of possible retrieval errors using the parameters in the OCO-2 retrieval correction. This approach

entails several steps. We first try to reproduce the OCO-2 observations using a regression. The predictor variables in this regression include seven different terrestrial biosphere model (TBM) estimates of net biome production and vegetation indices that have been input into an atmospheric transport model. We also include anthropogenic, ocean, and biomass burning flux estimates in the regression. A subsequent section (Sect. 2.3) describes this regression in greater detail. We save the model-data residuals from this regression. Subsequently, we regress these residuals on the parameters used in the OCO-2 bias correction.

These include aerosol optical depth and albedo, among other parameters (e.g., Wunch et al., 2011, see Sect. S1.4). Note that these retrieval parameters are not run through an atmospheric transport model, unlike the TBM fluxes. We estimate the retrieval errors as the portion of the model-observation residuals that are described by these retrieval parameters. The regression considers many different TBMs and vegetation indices, and it should therefore do reasonably well at reproducing patterns in the OCO-2 observations attributable to biospheric fluxes. Any remaining patterns in the data that map on to retrieval parameters

are likely due to retrieval errors rather than transport or flux errors.

We use a second approach to create an alternative set of simulated retrieval errors. Specifically, we model $XCO_2$ using fluxes from four different TBMs (Sect. 2.3) and compute the model-data residuals for each set of simulations. We identify the grid cells in which all four sets of residuals have the same sign (i.e., generated using four different TBMs) and identify all of the model grid cells in which the residuals have variable sign. In the former case, we take the median of all four residuals as the

estimated retrieval error and, in the latter case, assign a retrieval error of zero. This approach likely produces a conservative estimate of the retrievals errors (i.e., a possible underestimate); there is likely some amount of retrieval error at locations where we assign a retrieval error of zero.

## 2.3 Overview of the top-down experiments

We employ a top-down framework to evaluate the detectability of biospheric $CO_2$ fluxes using current OCO-2 observations.

Overall, we divide the globe into different hemispheres and biomes and determine whether we can detect flux patterns within each hemisphere or biome and each month. The term 'patterns' here refers to flux patterns that manifest at the resolution of an atmospheric model. We begin the analysis with very large hemispheric regions and then decrease the size of those regions until we are no longer able to detect any flux patterns beyond a mean $CO_2$ flux. That limit or end point is the smallest scale at which OCO-2 observations currently provide a constraint on $CO_2$ budgets. OCO-2 observations must be sufficient to detect more

than a mean flux across a region and month if future inverse modeling studies are to estimate biospheric $CO_2$ budgets at scales smaller than that region. Consequently, inverse modeling studies would generally be unable to obtain reliable information about the fluxes across smaller regions. This result bounds the type of information one can expect from the OCO-2 retrievals in their current stage of development. The remainder of this section explains this top-down analysis in greater detail.

5 We approach this problem using a regression framework. The regression attempts to reproduce OCO-2 observations of $XCO_2$ using predictor variables. These variables are the $XCO_2$ estimated by an atmospheric model; each model output estimates the $XCO_2$ enhancement due to fluxes in a particular region and a particular month, and we generate many model outputs using many different flux models (see Sect. 2.6). The regression has the following form:

$$z = h(\mathbf{X})\boldsymbol{\beta} + \boldsymbol{b} + \boldsymbol{\epsilon} \tag{1}$$

10 where $z$ are the OCO-2 observations (dimensions $n \times 1$). The matrix $\mathbf{X}$ ($m \times p$) has $p$ columns, and each column is a different $CO_2$ flux estimate for specific geographic region and a specific month. Each column of $\mathbf{X}$ has non-zero values for a specific region and month and zeros for all other regions and months. The function $h()$ is an atmospheric transport model (Sect. 2.6), and the model outputs, $h(\mathbf{X})$, have dimensions $n \times p$. The vector $\boldsymbol{b}$ ($n \times 1$) is the model spin-up or estimated $XCO_2$ at the beginning of the study time period, and $\boldsymbol{\epsilon}$ ($n \times 1$) are the regression residuals. Lastly, $\boldsymbol{\beta}$ ($p \times 1$) are the coefficients estimated as

15 part of the regression.

The regression provides a means to evaluate the detectability of biospheric $CO_2$ fluxes. At least some of the model outputs ($h(\mathbf{X})$) should describe substantial variability in the OCO-2 observations ($z$). Let's say that modeled $XCO_2$ using a particular flux model in a particular region and month help reproduce patterns in the OCO-2 observations. This result implies that OCO-2 observations are able to detect or constrain variability in $CO_2$ fluxes from that region and that month. By contrast, let's say

20 that no model outputs substantially improve the regression fit (i.e., no columns in $h(\mathbf{X})$). This result implies one of three things. First, the OCO-2 observations may not be sensitive to biospheric fluxes from that particular region and month. Second, there may be errors in current OCO-2 retrievals or in the atmospheric model that obscure surface fluxes from that region and month. Finally, all of the flux estimates used in $\mathbf{X}$ may be unskilled and may not match real-world conditions. We offer up a large number of flux models as possible predictor variables in $\mathbf{X}$, and at least some of these products should be expected

25 to correlate with real world $CO_2$ fluxes. In this case, it is unlikely that there is a shortage of reasonable $CO_2$ flux models to choose from. Rather, that result more likely reflects the sensitivity of the observations to surface fluxes, the maturity of current OCO-2 retrievals, or the accuracy of the atmospheric model. Hence, this approach provides a means to evaluate when and where current OCO-2 observations can constrain variability in biospheric $CO_2$ fluxes.

We utilize a model selection framework to determine which model outputs (i.e., columns of $h(\mathbf{X})$) describe substantial

30 variability in current OCO-2 observations. Model selection is a statistical approach common in regression modeling (e.g., Ramsey and Schafer, 2012, ch. 12). It will identify the set of predictor variables with the greatest power to describe the data. It also ensures that the regression does not overfit the data. The inclusion of more predictor variables in a regression will always improve model-data fit; a regression with $n$ independent predictor variables will always be able to describe $n$ data points perfectly. However, a model with $n$ independent predictor variables would overfit the data. For more on the dangers of

overfitting, refer to Zucchini (2000). To this end, one can use model selection to prevent overfitting and only include predictor variables that describe substantial variability in the observations.

We implement model selection based on the Bayesian Information Criterion (BIC) (Schwarz, 1978). We calculate a BIC score for many different combinations of predictor variables, and each combination has a different set of columns ($\mathbf{X}$). The best combination has the lowest BIC score:

$$BIC = L + p\ln(n^*) \tag{2}$$

where $L$ is the log likelihood of a particular combination of predictor variables (i.e., a particular configuration of $\mathbf{X}$). The log likelihood equation rewards combinations that improve model-data fit, and the second term in the equation ($p\ln n^*$) penalizes combinations with a greater number of predictor variables (i.e., columns in $\mathbf{X}$). This penalty also scales with the effective number of independent observations from OCO-2 ($n^*$, described in the SI), and it ensures that the selected model is not an over-fit.

A number of existing top-down studies of $CO_2$ use model selection (e.g., Gourdji et al., 2008, 2012; Shiga et al., 2014; Fang et al., 2014; Fang and Michalak, 2015; ASCENDS Ad Hoc Science Definition Team, 2015). Several utilize the approach to determine a set of environmental variables to include in a geostatistical inverse model (e.g., Gourdji et al., 2008, 2012). Other studies use model selection to determine whether existing $CO_2$ observations can constrain flux patterns from the biosphere (Fang et al., 2014) and from fossil fuel emissions (Shiga et al., 2014; ASCENDS Ad Hoc Science Definition Team, 2015). One study uses model selection to assess the capabilities of a proposed satellite mission (ASCENDS Ad Hoc Science Definition Team, 2015).

Model selection provides a convenient way to evaluate the information content of OCO-2 observations in their current state of development. One could also estimate $CO_2$ budgets in a Bayesian inverse model. The accuracy or uncertainty in those budgets would be indicative of the information content of the satellite observations. This approach can require a number of complex choices. In a Bayesian inverse model, one must choose a prior flux estimate, and there are many to choose from (e.g., Huntzinger et al., 2013; Peylin et al., 2013). An unskilled (or skilled) prior will have large (or small) prior uncertainties, and the posterior uncertainties will also be larger (or smaller) as a result. One could alias this effect for the capabilities of the observation network. A modeler must further differentiate between prior errors and errors in the retrieval/atmospheric model. Both can have complex, non-stationary structures (e.g., Liu et al., 2011). Inverse modeling with satellite observations also can be computationally intensive – both in terms of the number of atmospheric model simulations required and the computational requirements of the statistical inverse model. By contrast, the regression framework used here provides a simple metric to evaluate the information content of the observations. The statistical model implemented here does not require differentiating between different error types. Furthermore, we estimate the covariances locally (Sect. S1.3), making it easier and more computationally tractable to account for complex error structures.

## 2.4 Implementation of the top-down experiments

This section describes how the regression and model selection are implemented in the present study.

The regression begins with an intercept. The intercept is always included in the regression (in $\mathbf{X}$), and model selection can further add flux models to $\mathbf{X}$ to help reproduce the OCO-2 observations ($\mathbf{z}$). This intercept is a constant column of ones in $\mathbf{X}$. In the particular setup here, we include a different intercept for each region of the globe and each month. In other words, the intercept consists of multiple columns – one column for each region and month of the study period. This intercept is equivalent to a spatially and temporally constant flux in each region and month. Additional atmospheric model outputs $h(\mathbf{X})$ will not be selected unless they explain substantially more variability in the OCO-2 observations than this intercept or constant flux. The intercept plays an important role in the regression; it ensures that the regression will always be unbiased when averaged across the globe. Model selection could produce non-intuitive results if there were no intercept. Furthermore, intercepts are standard in regression modeling, and top-down, $CO_2$ studies that utilize model selection also include an intercept (e.g., Gourdji et al., 2008, 2012; Fang et al., 2014)

We subsequently run the regression with model selection three times to evaluate three different cases. In the first case, we divide the flux models into two large hemispheric regions, and we select different flux models for each hemispheric region and each month. The second and third experiments divide the fluxes into four continental regions and seven biomes, respectively (Fig. 2). The last experiment is more challenging than the first.

Note that we consider the model selection results from 2014 part of an initial model spin-up period and only report the results from 2015.

## 2.5 Synthetic data simulations

We subsequently utilize synthetic data simulations in this study to analyze the effects of different model or retrieval errors on the detectability of biospheric $CO_2$ fluxes. In these simulations, we create synthetic $XCO_2$ observations using an atmospheric transport model (Sect. 2.6) and the SiBCASA flux model. We then run model selection using these synthetic observations in place of real OCO-2 observations ($\mathbf{z}$). These model selection runs have the same setup as the real data simulations, except that the observations are synthetic instead of real. Furthermore, we only analyze the seven biome case in the synthetic data experiments. This case is more demanding of the observations than the two and four region cases; it is more difficult to obtain a robust constraint for seven regions than for two or four larger global regions. The seven biome case is also an important goal from a ecological perspective. For example, one might want to estimate how $CO_2$ fluxes differ in different tropical forests or in different temperate forests (e.g., on different continents or in different climate zones).

These synthetic simulations help to isolate the effect of different errors on the detectability of biospheric $CO_2$ fluxes. We first run the regression and model selection with no errors in the atmospheric model or in the retrievals ($\epsilon \approx \mathbf{0}$). We then successively add simulated error to the synthetic observations and evaluate how the model selection results change as the errors increase (Fig. 1). We include three different types of errors: flux errors, transport errors, and retrieval errors. The simulations with all errors included should produce model selection results similar to the real data experiments. Section 2.2 and the SI describe the simulated transport and retrieval errors.

The flux errors further account for plausible inaccuracies in the predictor variables within $\mathbf{X}$. No TBM or vegetation index has a distribution that perfectly matches real-world $CO_2$ fluxes. These errors affect our ability to detect biospheric fluxes using

the OCO-2 observations, and we therefore account for these flux errors in the synthetic data simulations. To this end, we remove SiBCASA as an option in the **X** matrix and choose among the other remaining flux models. This procedure simulates the plausible effects of imperfect flux models or predictor variables.

## 2.6 Atmospheric model simulations

We employ the Parameterized Chemistry Transport Model (PCTM) to model $XCO_2$ using a variety of surface flux models (Kawa et al., 2004). A number of existing studies use this model to simulate atmospheric $CO_2$ mixing ratios (e.g., Law et al., 2008; Gurney et al., 2009; Baker et al., 2010; Schuh et al., 2010; Shiga et al., 2013; ASCENDS Ad Hoc Science Definition Team, 2015; Hammerling et al., 2015). Several of these studies specifically use PCTM to model $CO_2$ in the context of satellite missions (e.g., Baker et al., 2010; ASCENDS Ad Hoc Science Definition Team, 2015; Hammerling et al., 2015). The PCTM
configuration in this study has global coverage, a spatial resolution of 1° latitude by 1.25° longitude, and 56 vertical levels. We both input the fluxes and estimate atmospheric mixing ratios at a 3-hourly time resolution, and the model transports fluxes through the atmosphere using winds from NASA's Modern-Era Retrospective Analysis for Research and Applications (MERRA) (Rienecker et al., 2011). Section S1.1 includes more detail on the model initial condition and spin-up period.

We subsequently model $XCO_2$ using several different terrestrial biosphere models (TBMs) and vegetation indices, and
these model outputs are incorporated into model selection ($h(\mathbf{X})$). We include four TBMs with contrasting spatial features from MsTMIP, the Multi-scale Synthesis and Terrestrial Model Intercomparison Project (Huntzinger et al., 2013; Fisher et al., 2016a, b). Section S1.2 describes MsTMIP and the TBMs in greater detail. We also include SIF (solar-induced fluorescence) from the Global Ozone Monitoring Experiment-2 (GOME-2, Joiner et al., 2013) as well as EVI (enhanced vegetation index) and NDVI (normalized difference vegetation index) from the Moderate-Resolution Imaging Spectroradiometer (MODIS; e.g.,
Huete et al., 2002). Note that we directly input these vegetation indices into PCTM as a surface 'flux.' The regression will adjust the magnitude of the transport model outputs to reproduce the OCO-2 observations, so the absolute magnitude of the vegetation indices is not important. Rather, we are interested in whether the patterns in these vegetation indices help reproduce patterns in the OCO-2 observations, potentially in combination with other indices or TBMs.

We also consider non-biospheric fluxes for use in **X**. We include anthropogenic emissions from EDGAR v4.2 FT2010
(European Commission, Joint Research Centre (JRC)/Netherlands Environmental Assessment Agency (PBL), 2013; Olivier et al., 2014), climatological ocean fluxes from Takahashi et al. (2016), and biomass burning fluxes from the Global Fire Emissions Database (GFED), version 4.1 (van der Werf et al., 2010; Giglio et al., 2013). We are not interested in anthropogenic or marine fluxes per se. Rather, we want to account for these fluxes in the modeling framework and do not want any omissions to affect inferences related to biospheric fluxes. As a result, we do not separate these non-biospheric fluxes by region or month
because these sources are not the focus of this study; each of these sources is assigned a single column in **X**. Furthermore, we do not include these variables by default within **X**, unlike the constant flux base model. Rather, they are included as candidate variables within the model selection framework.

Note that in the seven biome region case, **X** has a minimum of 112 columns and a maximum of 899 columns. 112 columns associated with the intercept and are always included in **X** (i.e., 16 months × 7 biomes per month). 784 columns associated

with biospheric fluxes (7 flux models $\times$ 16 months $\times$ 7 biomes), and three columns associated with fossil fuel, ocean, and biomass burning fluxes. These columns may or may not be included in $\mathbf{X}$, depending upon the results of model selection.

## 3   Results & discussion

### 3.1   The biospheric $CO_2$ signal versus model and retrieval errors

We compare simulated model and retrieval errors against the atmospheric signal from biospheric fluxes (Fig. 3). The comparison provides an intuition of the errors and the $CO_2$ 'signal' given current modeling and observation capabilities. Prior to the OCO-2 satellite launch, several studies modeled the $XCO_2$ signal from surface fluxes (e.g., Olsen and Randerson, 2004), and the measurement precision required for space-based constraints on $CO_2$ fluxes (e.g., Rayner and O'Brien, 2001). It is now possible to make this evaluation with real instead of synthetic observations. Also note that the $XCO_2$ signal and estimated

errors will vary depending on the averaging time period. We report monthly averages of the biospheric signal and the errors. Many top-down, inverse modeling studies report monthly flux totals, so all of the analysis presented here is aggregated to one month averages.

The atmospheric $CO_2$ signal from biospheric fluxes is marked, even when averaged across a total vertical column (Fig. 3)– a global mean absolute value of 0.5ppm in February and 1.3ppm in July. The 10th and 90th percentiles are 0.04 and 1.4ppm in

February and 0.06 and 3.8ppm in July. In July, the largest enhancements are in the northern hemisphere mid and high latitudes while the largest enhancements during winter months are in the tropics and southern hemisphere.

By contrast, the simulated model and retrieval errors are comparable to this $XCO_2$ signal from biospheric fluxes. These errors have a mean absolute value of 0.6ppm in both February and July. The 10th and 90th percentiles are 0.08 and 1.35ppm in February and 0.08 to 1.25ppm in July. Using the alternative retrieval estimate, the errors and percentiles are 0.8, 0.02, and

2.1ppm in February and 1.4, 0.05, and 3.7ppm in July. Phrased differently, these errors, averaged across all nadir and target data, equate to 115 – 122% of the mean biospheric $CO_2$ signal in February and 43 – 107% in July, depending upon the retrieval error estimate.

It is important to note that the distribution of these observations is heterogeneous across the globe (Fig. 3), even though the total number of observations is large (e.g., 268,671 and 343,053 nadir and target observations in February and July, respectively,

in the lite data file). For example, the data are concentrated in tropical and temperate regions and are sparse at high latitudes and regions with frequent cloud cover (e.g., the Amazon).

These errors are not inconsistent with those estimated by existing studies. Wunch et al. (2017) compare OCO-2 observations against TCCON observations and report an average site bias of 0.22ppm when using land nadir retrievals and a root mean squared error of 1.31ppm. Furthermore, Worden et al. (2017) estimate a precision and accuracy, respectively, of 0.75 and

0.65ppm for land nadir observations. These studies are not necessarily directly comparable because each uses different metrics and observations. Furthermore, several compare OCO-2 to TCCON, the same observations used to bias correct OCO-2. With that said, the overall numbers reported by different studies are not inconsistent with one another and with Fig. 3.

The relative magnitude of the errors provides an informative measure of the observations, but it does not tell the complete story. A number of other considerations affect scientists' ability to estimate surface fluxes using these observations. First, inverse models leverage more than the point-wise signal to estimate surface fluxes; these models leverage complex spatial and temporal patterns in the data to estimate surface fluxes. Second, the absolute magnitude or variance of the errors is only one consideration. Another important factor is the spatial and temporal correlations or covariances in these errors. These covariances reduce the independent information in the data and can obscure patterns in $XCO_2$ that are due to surface fluxes. As a result, we construct real and synthetic data experiments to understand what these errors mean for estimating surface fluxes.

## 3.2 Real data experiments

The model selection experiments using real data indicate the number or size of regions for which we can reliably constrain biospheric $CO_2$ budgets using current OCO-2 observations. We start the real data simulations with large hemispheric regions and reduce the size of the regions until we are no longer able to detect any $CO_2$ flux patterns or information beyond a mean monthly flux in each region and each month. We would need to detect more than a mean flux from a given region if we are to reliably constrain fluxes across smaller regions.

The first real data experiment indicates whether the observations are sufficient to detect flux patterns within two large hemispheric regions (Fig. 2). Figure 4a displays the number of months in which at least one model output (i.e., column of $\mathbf{X}$, Eq. 1) is chosen, broken down by region. The results in Fig. 4a are grouped by season for convenience.

Model selection identifies flux patterns in about half of all months. This outcome suggests that OCO-2 and the PCTM model can be used to identify broad, hemispheric flux patterns. One important exception is the extra-tropics (e.g., the temperate, boreal, and arctic region), in both spring and fall. Biospheric uptake in these seasons is less than the summer maximum, in both the northern and southern hemispheres. As a result, flux patterns in these areas are not as heterogeneous and not readily detectable using the satellite observations. This result also indicates that the OCO-2 observations can be used to reliably constrain $CO_2$ budgets at scales smaller than hemispheric in about half of all cases. With that said, we do not select a single flux pattern in three of four seasons for one hemisphere.

In a second experiment, we try to identify flux patterns within four, smaller regions using OCO-2 observations and model selection (Fig. 4b). At least one flux pattern is selected in 29% of all regions and months, and this result suggests that inverse modeling studies would be able to constrain $CO_2$ budgets at more detailed spatial scales in about one third of all regions and months. This experiment is more demanding than the first, and it is therefore unsurprising that fewer flux patterns are selected. Flux patterns within these four continental regions are often less heterogeneous than across the two large hemispheric regions in the first experiment.

The third and final experiment includes seven biomes, and relatively few flux patterns are selected in this final experiment (16% of all possible regions and months, Fig. 4c). This result suggests a limited ability to detect biospheric flux patterns within each of the seven global biomes. Inverse modeling studies would thus be able to uniquely constrain $CO_2$ budgets across smaller regions in only a small handful of cases (e.g., 16% of all possible regions and months). These results appear similar to a recent study using current GOSAT retrievals. Houweling et al. (2015) compare an ensemble of inverse modeling flux estimate using

GOSAT. Estimates show good agreement across very large regions (e.g., within 20% for global, annual $CO_2$ budgets) but disagree by over 100% over subcontinental-sized TransCom regions (e.g., Gurney et al., 2002).

As part of model selection, we also evaluate the effective number of independent OCO-2 observations ($n^*$, Sect. S1.3), and we estimate one independent observation per ∼1200 OCO-2 lite retrievals. Correlations or covariances in transport and retrieval errors will reduce the value of $n^*$. The estimate is similar among all of the real data experiments and is approximately 4000 (4060 for the two region case, 3600 for the four region case, and 3540 for the seven region case). By comparison, the total number of OCO-2 observations during the study period ($n$) is $5.08 \times 10^6$. Note that the value of $n^*$ decreases as the number of regions increases. Fewer and fewer model outputs are selected as the number of regions increases. As a result, there are more residual, unexplained flux patterns in the seven region case than in the two region case. The regression residuals have larger covariances in the seven region case, and the observations become less independent.

These results are broadly consistent with the OCO-2 science team's ongoing flux inter-comparison study (Crowell , 2017). Several research groups are developing inverse models to estimate $CO_2$ fluxes using OCO-2 observations, and a comparison of these estimates provides insight into the robustness of the flux estimates. To date, these comparisons often show relatively good agreement for total hemispheric terrestrial budgets, but the ensemble of estimates diverges for smaller regions. Other, newly published inverse modeling studies use OCO-2 observations to estimate regional budgets (e.g., Liu et al., 2017). These studies primarily focus on questions about carbon cycle science, and many employ frameworks that are not necessarily intended to exhaustively sample all plausible sources of uncertainty.

### 3.3 Synthetic data experiments

The goal of the synthetic data simulations is to understand the challenges that influence the real data results. If future efforts can mitigate these challenges, then inverse modeling studies would be able to reliably constrain flux patterns and $CO_2$ budgets across individual biomes or even smaller regions.

We first construct an idealized synthetic data study without any errors (Fig. 5a). This case study indicates that the OCO-2 observations are not inherently insensitive to biospheric fluxes at the surface, a result consistent with previous studies (e.g., Olsen and Randerson, 2004). In this idealized case, a flux pattern is selected for every biome in every month using model selection.

Subsequent model selection experiments include at least one error type, and flux patterns are selected in fewer regions and months in all of these cases (Fig. 5b-d). Of the three different error types, retrieval errors have the largest impact on the model selection results (Fig. 5d). Note that the retrieval errors used to generate Fig. 5d are those simulated in Fig. 1c-d. Section S2.2 describes the model selection experiments using an alternative set of retrieval errors (Fig. 1e-f), and these results are similar to those presented in the main manuscript (Fig. 5).

Notably, the addition of any error, large or small, appears to hinder flux pattern detection in marginal biomes – biomes with small fluxes and/or small spatial and temporal variability (e.g., tundra and deserts). Arguably, this result is unsurprising. OCO-2 observations are sparse in many cloudy, high latitude regions, and $CO_2$ fluxes are weak at high latitudes and in deserts. Fluxes from these regions are quickly obscured by even modest errors in the model or observations. Future $CO_2$ remote sensing efforts

would have difficulty detecting biospheric patterns within these areas. Other regions, like forests and grasslands, have larger and/or more heterogeneous fluxes, and these patterns should be easier to detect with satellite observations.

One of the experiments specifically accounts for errors in existing flux estimates (Fig. 5b). We find that these errors do have an effect on the results, but that effect is not nearly as large as that due to retrieval errors. We argue that current OCO-2 observations can detect patterns in $CO_2$ fluxes if at least one model output helps explain substantial variability in those observations. We offer up a number of different model outputs using several different flux models, but there is always a possibility that none of these flux models adequately approximates real world fluxes. The experiment in Fig. 5b evaluates how these flux errors could affect the result. Furthermore, if there are large, unresolved patterns in anthropogenic or marine fluxes, these unresolved patterns could also influence the results for biospheric fluxes. Note that this set of issues also affects Bayesian inverse models and therefore has implications beyond the methodology used in this study. Specifically, the availability and skill of the prior flux estimate affects the robustness and uncertainty of the inverse modeling estimate.

Subsequent experiments (Fig. 5e-g) include two different error types, and flux patterns are selected in fewer months and regions relative to the previous cases. The experiment with transport and flux errors (but not retrieval errors, Fig. 5e) still show good detectability during the summer months and in temperate and tropical forest and grassland biomes. By contrast, the experiments that include retrieval errors (Fig. 5f-g) show limited detectability in all biomes and seasons.

The last experiment (Fig. 5h) includes all error types, and we obtain positive results in fewer regions in fewer months relative to other cases. These results are broadly consistent with the real data experiments; a similar number of regions and seasons are selected in both this experiment and the real data experiment (Fig. 5h). This consistency indicates that the synthetic simulations likely mirror real-world conditions. Note that the estimate for $n^*$ in this final synthetic experiment is about half that of $n^*$ in the real data experiments ($n^* = 1630$). These synthetic experiments may therefore slightly overestimate the error correlations or covariances and underestimate the variance or white noise portion of the errors.

Overall, the synthetic simulations suggest that retrieval errors play a salient role relative to other error types (e.g., transport errors or flux errors, Fig. 5). Spatial and temporal error covariances and biases may be at least partly to blame. The estimated transport errors are spatially and temporally correlated on synoptic time scales (e.g., Liu et al., 2011; Miller et al., 2015). These scales are generally smaller than the biomes and hemispheres examined in this study. As a result, these errors will average down over time and space, and this averaging will mitigate the impact of these errors on the results. This statement, of course, only holds if there are no large-scale biases in the meteorology. The simulated retrieval errors, by comparison, covary across longer spatial and temporal scales. These errors correlate with retrieval parameters like aerosol optical depth or albedo that often change at broader seasonal or regional scales. The greater these error correlations, the less these errors average down across space and time, and the greater impact these errors have on the utility of $XCO_2$ observations. A reduction in the spatial and temporal coherence of these errors would improve the model selection results.

## 4  Conclusions

The OCO-2 satellite offers a new, global window into atmospheric $CO_2$ and $CO_2$ fluxes at the Earth's surface. This study explores a first step in realizing these capabilities; we evaluate the extent to which current OCO-2 observations can detect patterns in biospheric $CO_2$ fluxes and constrain monthly $CO_2$ budgets.

5    We find that OCO-2 observations, in their current state of development, often provide a reliable constraint on $CO_2$ budgets across continental or hemispheric regions. By contrast, we find that current observations can provide a unique $CO_2$ estimate across smaller regions in only a handful of cases. As a result, inverse modeling studies are unlikely to constrain regional fluxes at fine spatial and temporal scales given the current maturity of the observations. Regional $CO_2$ budgets estimated using these observations would be highly uncertain and prone to biases.

10   These results do not reflect any inherent limitation in the sensitivity of the OCO-2 satellite. Rather, a set of synthetic data simulations indicate that these limitations are likely the result of large errors: retrieval errors and to a lesser extent atmospheric transport errors (Fig. 5). Hence, the value or potential of the OCO-2 observations is greater than these results might otherwise imply. For example, the retrieval errors simulated in this study often covary across large regions and across a month or more. Future improvements to retrieval algorithms could reduce both the variance and covariance of these errors, enabling confident 15   $CO_2$ flux constraints across smaller regions.

Even with these limitations, current OCO-2 observations provide new information on $CO_2$ fluxes for many regions of the globe. On one hand, in situ data appear to provide a stronger constraint on $CO_2$ fluxes in some well-instrumented regions of the world, like North America (e.g., Fang et al., 2014). Results using seven global biomes show only a limited ability of current OCO-2 observations to differentiate among regions. On the other hand, in situ observations are sparse in many regions 20   of the world, including in most of the tropics, Africa, South America, and Asia (e.g., NOAA Global Monitoring Division, 2017). Current OCO-2 observations bring new monitoring capabilities to these regions that are unlikely to be matched by in situ observations within the near future.

Furthermore, a number of new satellite missions will launch in the next five years. Multiple sets of observations, in tandem, will provide a more detailed, robust constraint on $CO_2$ fluxes. For example, the GOSAT-2 satellite will monitor atmospheric 25   $CO_2$ with better accuracy relative to the original GOSAT satellite (Japan Aerospace Exploration Agency, 2017). This improvement in both the quality and overall quantity of $CO_2$ observations will enable more detailed estimates of $CO_2$ fluxes. In addition to GOSAT-2, the OCO-3 mission will observe $CO_2$ from the International Space Station at a different locations and times of day relative to OCO-2 (NASA Jet Propulsion Laboratory, 2017). This feature will provide a stronger constraint on spatial and temporal variations in $CO_2$ fluxes. However, retrieval errors appear to be a key factor in our results and will likely 30   be a challenge for these future missions. Work on the OCO-2 retrieval algorithm will inform these upcoming missions, so improvements to the OCO-2 retrievals will likely improve the data capabilities of future missions as well. Further improvements to the satellite retrieval and atmospheric transport modeling could enable OCO-2 and future missions to provide detailed $CO_2$ budgets for much finer regions.

*Competing interests.* The authors declare that they have no conflict of interest.

*Acknowledgements.* We thank David Baker, David Crisp, Benjamin Poulter, and Eva Sinha for their help and feedback on the manuscript. The OCO-2 data are produced by the OCO-2 project at the Jet Propulsion Laboratory, California Institute of Technology, and obtained from the OCO-2 data archive maintained at the NASA Goddard Earth Science Data and Information Services Center. OCO-2 data are

5   publicly available online at https://co2.jpl.nasa.gov/#mission=OCO-2. The MODIS EVI and NDVI data products are available online at https://lpdaac.usgs.gov/ courtesy of the NASA Land Processes Distributed Active Archive Center (LP DAAC), USGS/Earth Resources Observation and Science (EROS) Center, Sioux Falls, South Dakota. GOME-2 SIF data are available at https://acd-ext.gsfc.nasa.gov/People/Joiner/my_gifs/GOME_F/GOME-F.htm courtesy of Joanna Joiner. MsTMIP data products are archived at the ORNL DAAC (http://daac.ornl.gov), and 3-hourly flux products are available at dx.doi.org/10.3334/ORNLDAAC/1315. This work is funded by the Carnegie Distinguished Post-

10  doctoral Fellowship and NASA grant #NNX13AC48G. Funding for MsTMIP was provided through NASA ROSES Grant #NNX10AG01A. Data management support for preparing, documenting, and distributing model driver and output data was performed by the Modeling and Synthesis Thematic Data Center at Oak Ridge National Laboratory (ORNL; http://nacp.ornl.gov), with funding through NASA ROSES Grant #NNH10AN681.

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

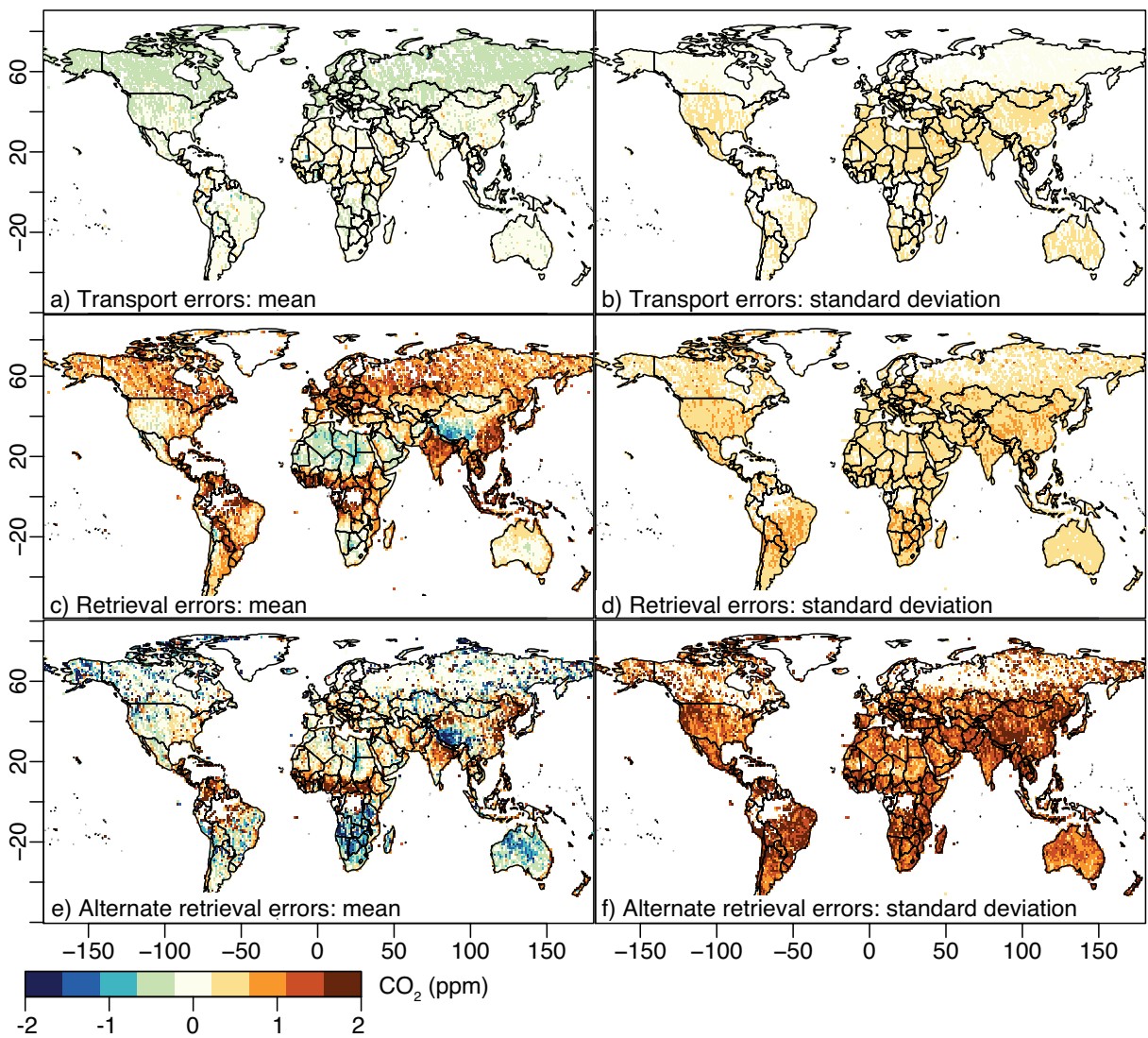

**Figure 1.** This figure displays the simulated synthetic data errors (i.e., simulated $\epsilon$): the simulated atmospheric transport errors (a-b), the simulated observation/retrieval errors (c-d), and a second, alternative set of observation/retrieval errors (e-f). The left-hand panels display the mean of the errors within each PCTM grid box across the entire 2014–2015 study period, an indication of the correlation or covariance among the errors. By contrast, the right-hand panels display the standard deviation of the errors or residuals within each PCTM grid box.

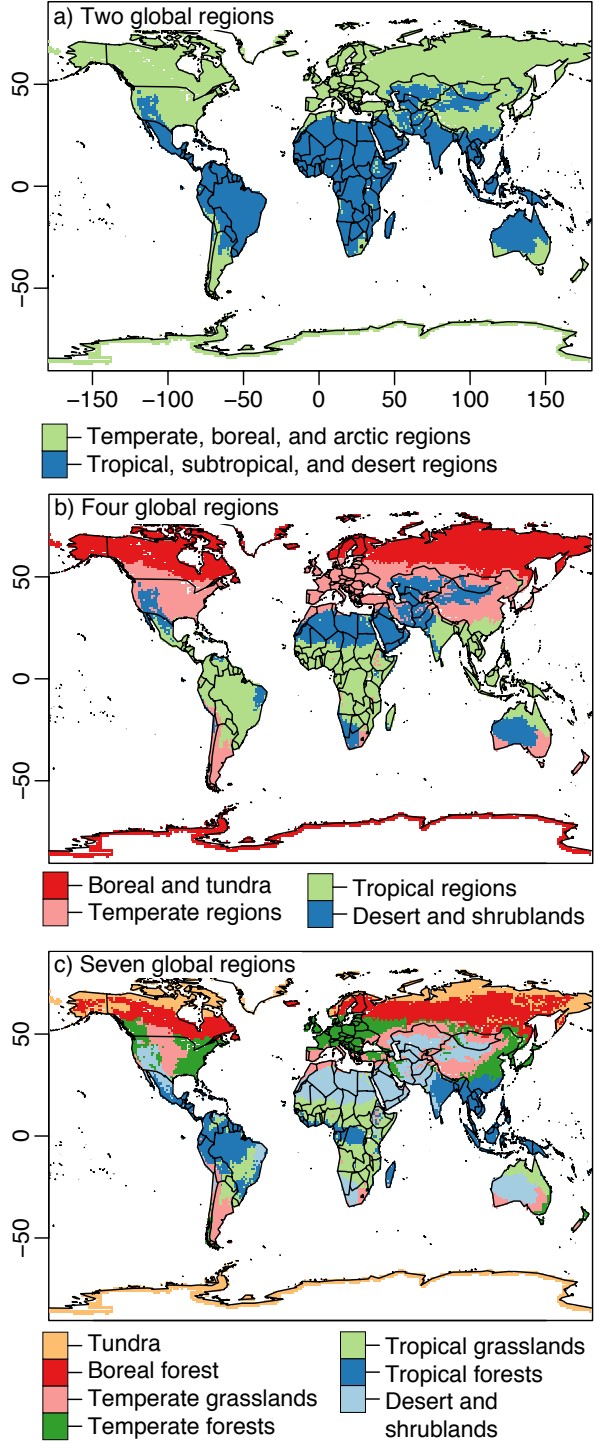

**Figure 2.** The two hemispheric regions (a), four continental regions (b), and seven biomes (c) used in this study. The biomes are based on a world biome map by Olson et al. (2001). The two and four region maps are amalgamated versions of the biomes.

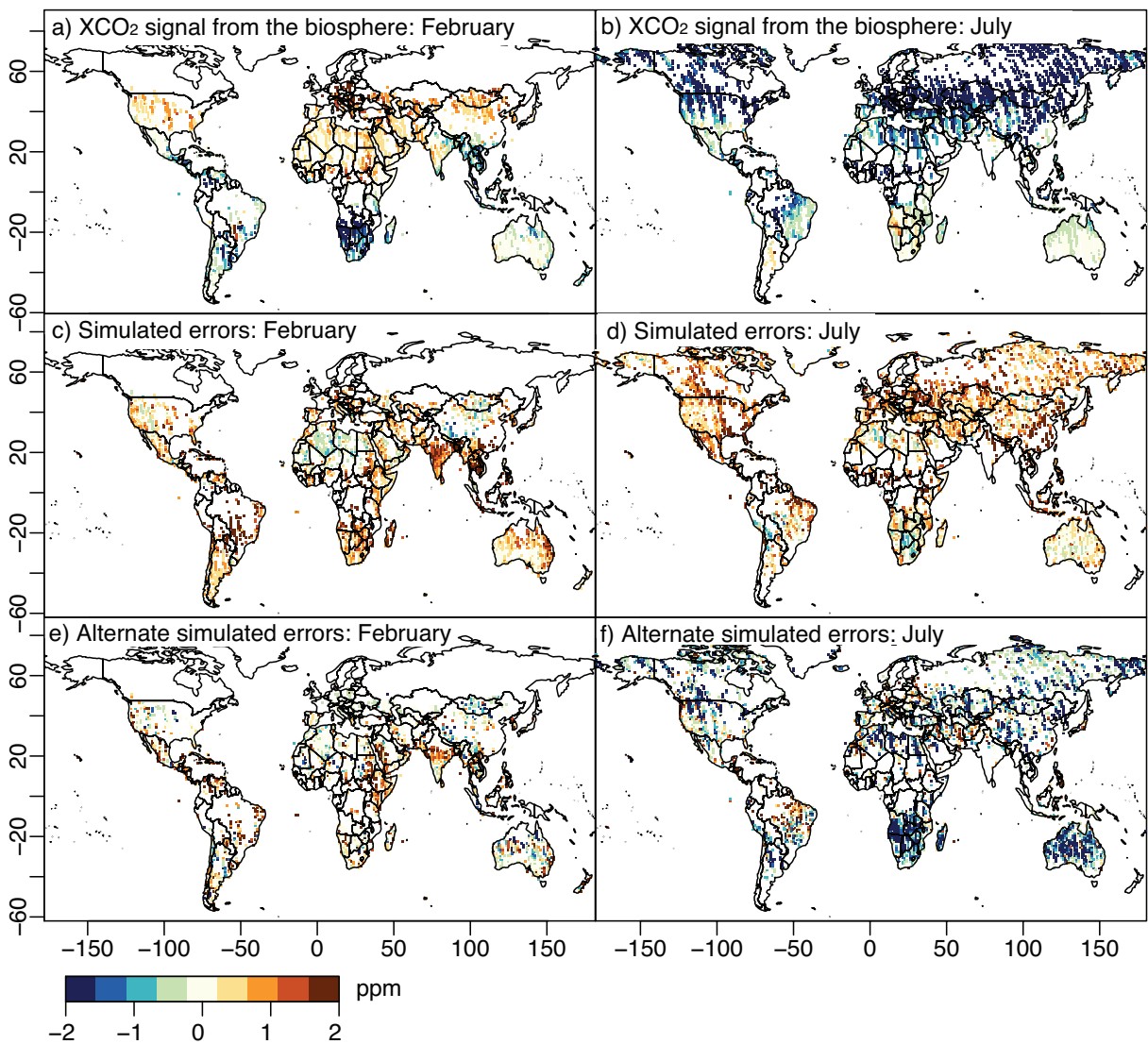

**Figure 3.** This figure compares the total column $CO_2$ or $XCO_2$ signal from biospheric fluxes against simulated model and observation errors. Panels a and b display the monthlong mean $XCO_2$ signal from biospheric $CO_2$ fluxes for February and July, respectively. We estimate this signal using the SiBCASA flux model and PCTM. Also note that the $XCO_2$ signal for February and July includes $CO_2$ fluxes from the months of February and July, respectively, and no fluxes from previous or subsequent months. Panels c and d represent the sum of both simulated observation and atmospheric transport errors (monthlong mean, Sect. 2.2). Panels e and f show the sum of these errors using an alternate estimate for the retrieval errors.

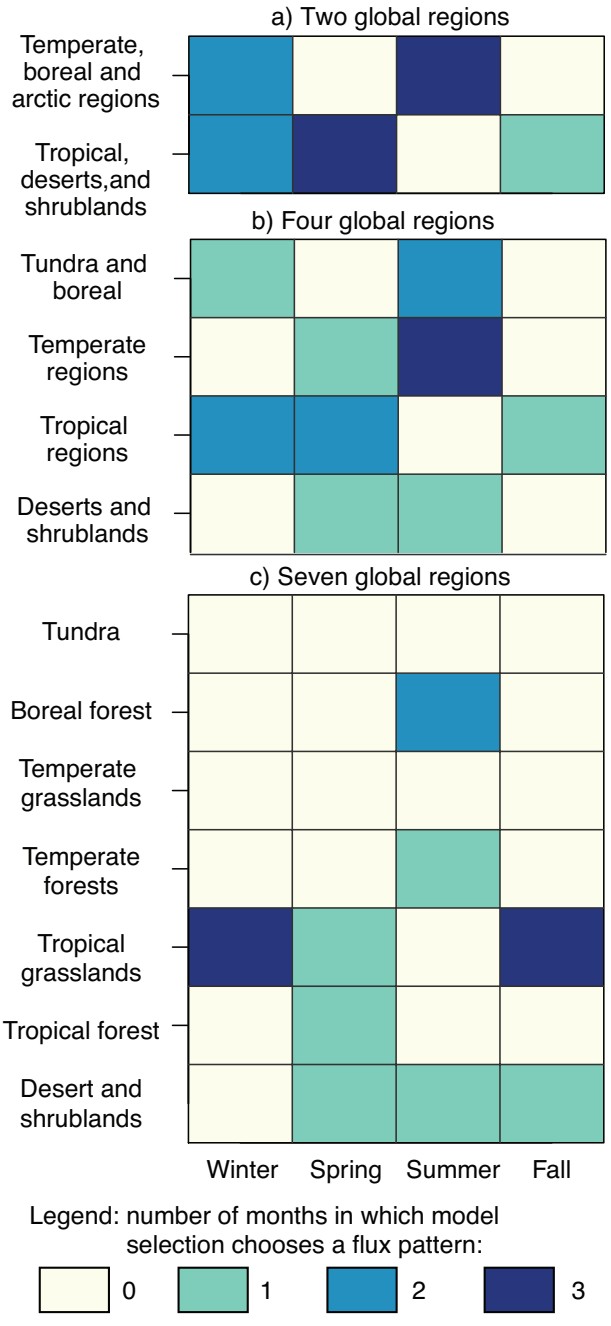

**Figure 4.** The results of the model selection experiments using real data. The different colors indicate the number of months in which at least one flux pattern is selected for a given region, and dark colors suggest excellent detectability while light colors suggest limited detection abilities. Flux patterns are selected for a greater fraction of regions/months in the two region case (a) than in the four or seven region cases (b and c).

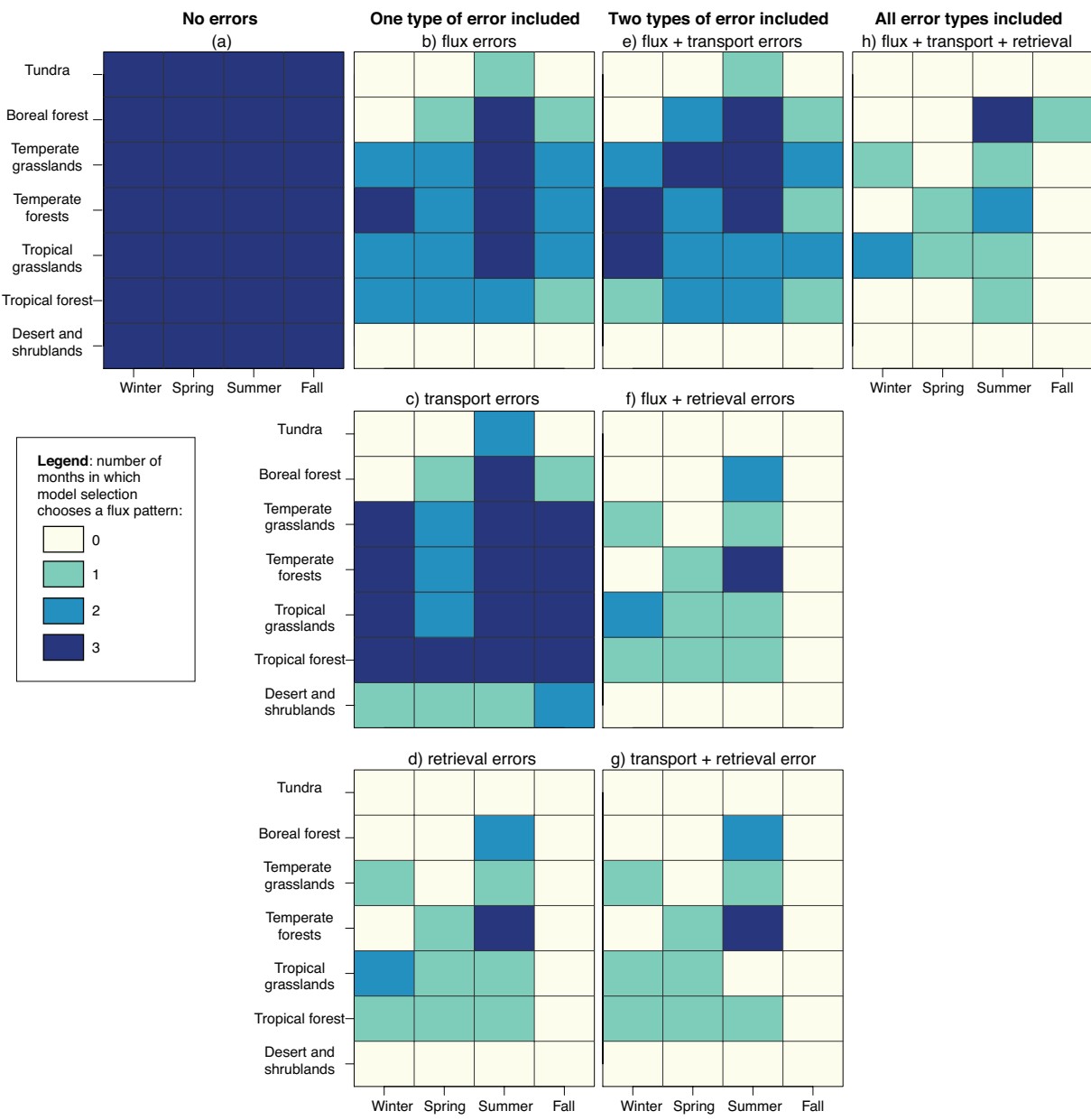

**Figure 5.** Model selection results for the synthetic data experiments. The first column (a) shows an experiment with no errors in the synthetic observations. The results of that experiment are ideal, and at least one flux pattern is selected in every month and every region using model selection. Subsequent panels (b-h) show the results with one, two, and three types of errors included. Fewer regions and seasons are selected in these experiments. The plot signals that retrieval errors likely play a key role.