# Peer review of "Characterizing biospheric carbon balance using $CO_2$ observations from the OCO-2 satellite"

_Atmospheric Chemistry and Physics, 2017_

## Referee Comment (RC1) · Anonymous Referee #1 · 31 Oct 2017

The authors analyze the information content of the OCO-2 XCO$_2$ retrievals in terms of surface fluxes. They first look for typical patterns originating from surface fluxes in real XCO$_2$ measurements and then use a simulation framework to document this performance. I found it particularly difficult to follow the logic of the paper and to evaluate the soundness of the approach. As a preliminary step for publication, the authors should seriously invest in making their study accessible to the broad audience of ACP. As a second step, I would like to highlight the following issues.

- The paper concludes to a limited utility of OCO-2 retrievals for flux estimation with current retrieval algorithms and transport model. This may be correct, but is

orthogonal to the claim made by Liu et al. (2017). The disagreement should be clearly stated.

- Section 3.1 and the first part of Section 3.3 reinvent the wheel. See, e.g., Olsen and Randerson (2004) and Worden et al. (2017). Similarly, l. 23-28 are just an adaptation of an old argument (Rayner and O'Brien, 2001).

- The retrieval error simulations of Fig. 3 look overly optimistic in comparison to the validation results of Wunch et al. (2017).

- Section 3.2 looks for flux patterns in $XCO_2$. Most top-down studies from OCO-2 would use a Bayesian approach where flux-error patterns are looked for. This is more challenging because the signal is even smaller (while the paragraph in-between p. 5 and p. 6 suggests that the two approaches are rather equivalent with respect to the measurement information content). One should therefore discuss this limitation and further tone down the conclusions of the paper.

**References**

Liu, J. et al. Contrasting carbon cycle responses of the tropical continents to the 2015–2016 El Niño. Science 358, eaam5690 (2017). Olsen, S. C., and J. T. Randerson (2004), Differences between surface and column atmospheric CO2 and implications for carbon cycle research, J. Geophys. Res., 109, D02301, doi:10.1029/2003JD003968.

Rayner, P. J. and O'Brien, D. M.: The utility of remotely sensed CO2 concentration data in surface source inversions, Geophys. Res. Lett., 28, 175–178, 2001.

Worden, J. R., Doran, G., Kulawik, S., Eldering, A., Crisp, D., Frankenberg, C., O'Dell, C., and Bowman, K.: Evaluation and attribution of OCO-2 XCO2 uncertainties, Atmos. Meas. Tech., 10, 2759-2771, https://doi.org/10.5194/amt-10-2759-2017, 2017.

[Figure]

Wunch, D., Wennberg, P. O., Osterman, G., Fisher, B., Naylor, B., Roehl, C. M., O'Dell, C., Mandrake, L., Viatte, C., Kiel, M., Griffith, D. W. T., Deutscher, N. M., Velazco, V. A., Notholt, J., Warneke, T., Petri, C., De Maziere, M., Sha, M. K., Sussmann, R., Rettinger, M., Pollard, D., Robinson, J., Morino, I., Uchino, O., Hase, F., Blumenstock, T., Feist, D. G., Arnold, S. G., Strong, K., Mendonca, J., Kivi, R., Heikkinen, P., Iraci, L., Podolske, J., Hillyard, P. W., Kawakami, S., Dubey, M. K., Parker, H. A., Sepulveda, E., García, O. E., Te, Y., Jeseck, P., Gunson, M. R., Crisp, D., and Eldering, A.: Comparisons of the Orbiting Carbon Observatory-2 (OCO-2) XCO2 measurements with TCCON, Atmos. Meas. Tech., 10, 2209-2238, https://doi.org/10.5194/amt-10-2209-2017, 2017.

---

## Author Comment (AC1) · 10 Nov 2017

We would like to thank the reviewer for ideas and suggestions for the manuscript. This feedback will be very helpful for updating and improving the manuscript. Below, we have included both the reviewer's suggestions (in bold) along with the associated changes we plan to make.

- **I found it particularly difficult to follow the logic of the paper and to evaluate the soundness of the approach. As a preliminary step for publication, the authors should seriously invest in making their study accessible to the**

[Figure]

**broad audience of ACP.**

This is very helpful feedback for framing the paper and describing the components of the methodology. We plan to re-frame the paper logic in several ways to make it more accessible to a broad audience. First, we plan to expand the overall description of the paper and description of the general approach at the end of the introduction (pg. 2, line 28 to pg. 3, line 21). We will describe the paper narrative in non-technical terms to give the reader an intuitive, high-level overview of the paper logic and flow. This description would provide better intuition for a wide audience of readers, especially those readers who may skip over the more technical information in the methodology (sect. 2).

Second, we will simplify the methods section (sect. 2) so that it is accessible to a broad audience. For example, this section contains seven equations. We will move several of these equations to the SI (e.g., Eq. 4-7) and instead expand the non-technical portions of the description. In this way, the paper will still include all of the technical detail for readers who want it, but the description in the main paper will be accessible to a broader audience.

Third, we will provide more references to existing studies that use similar approaches. Readers who are interested in more details on the methodology could gain greater context using these references. We will make this change throughout the manuscript and particularly from pg. 6, line 10 to pg. 7, line 20.

- **The paper concludes to a limited utility of OCO-2 retrievals for flux estimation with current retrieval algorithms and transport model. This may be correct, but is orthogonal to the claim made by Liu et al. (2017). The disagreement should be clearly stated.**

We will discuss this difference in the revised version of the manuscript. Liu et al. (2017) was published after this ACPD manuscript, so it is only now possible to make this comparison. Liu et al. use an atmospheric inversion to estimate

$CO_2$ fluxes for different tropical regions of the globe. They estimate uncertainties in their regional budget estimates, and those uncertainties are generally smaller than implied by the current ACPD manuscript.

Liu et al. (2017) use a 4DVAR approach and estimate the posterior uncertainties using a small number of Monte Carlo simulations. However, these uncertainty estimates are likely to be underestimates – due to compromises required to make the inversion computationally tractable. For example, most satellite-based inversions like Liu et al. (2017) do not fully account for error correlations or biases in the observations and atmospheric model; these studies typically use a diagonal error covariance matrix. Furthermore, Liu et al. (2017) and other studies use a small number of Monte Carlo simulations to estimate the errors (e.g., 60 simulations in Liu et al. (2014)). By contrast, Ribgy et al. (2011) and Ganesan et al. (2014) argue that 100,000 and 25,000 realizations are necessary to robustly estimate uncertainties for their particular inverse modeling problems. Note that it is not always possible to generate large numbers of realizations or fully account for error correlations in current satellite-based inverse models due to computational constraints. In the ACPD manuscript, we do not use a 4DVAR inverse model for this reason.

Consistent with this interpretation, results from the OCO-2 flux team ongoing intercomparison study indicate much larger uncertainties in estimated fluxes. The results are broadly consistent with those presented in the current ACPD manuscript (e.g., Crowell et al. 2017); preliminary results indicate that OCO-2 observations currently provide robust constraints for hemispheric regions but provide weaker constraints for individual continents or subcontinents. More specifically, recent flux team comparisons include $CO_2$ flux estimates from about eight different inverse modeling groups, and the level of disagreement among these estimates provides a measure of uncertainty in current top-down flux estimates that use the same version of the OCO-2 retrievals as applied in the current work

and in Liu et al. (2017). These estimates (using nadir observations) often show relatively good agreement for total hemispheric terrestrial $CO_2$ budgets, with the disagreement among inverse modeling estimates being smaller than the total $CO_2$ budget for a given hemisphere. The opposite is often true of $CO_2$ budgets estimated for smaller regions (e.g., Sub-Saharan Africa or Tropical Asia), with the disagreement among inverse modeling estimates usually being larger than the total budget. This ongoing work is consistent with the interpretation in the current manuscript.

- **Section 3.1 and the first part of Section 3.3 reinvent the wheel. See, e.g., Olsen and Randerson (2004) and Worden et al. (2017). Similarly, l. 23-28 are just an adaptation of an old argument (Rayner and O'Brien, 2001).**

The studies mentioned above investigate several requirements for constraining carbon budgets with satellite observations. Rayner and O'Brien (2001) explore the measurement precision required for space-based constraints on surface $CO_2$ fluxes. Olsen and Randerson (2004) model $XCO_2$ column enhancements across the globe due to surface $CO_2$ fluxes and compare them with surface enhancements. Lastly, Worden et al. (2017) estimate the errors in OCO-2 $XCO_2$ observations.

As the reviewer points out, the concepts used in Sects. 3.1 and 3.3 are, in part, built on these earlier approaches. However, the purpose of this section is not to develop new concepts. Rather, we build on existing concepts to assess real OCO-2 data. Rayner and O'Brien (2001) and Olsen and Randerson (2004), by contrast, did not have any real $XCO_2$ observations at their disposal, only simulations of possible future observations. Furthermore, we feel that these sections provide useful context and improve the manuscript narrative. Much of the manuscript presents the results of statistical experiments. These experiments use, as inputs, $XCO_2$ observations from OCO-2 and estimates of atmospheric transport and satellite retrieval errors. Sect. 3.1 provides visualizations of those

inputs.

In the revised manuscript, we will cite the studies listed above, clarify that we use concepts from the studies, and explain that we apply those concepts to real observations from OCO-2. We will also compare the retrieval errors in Sect. 3.1 against those in Worden et al. (2017). Lastly, we will shorten the first part of Sect. 3.3. That section presents the synthetic study results with no errors; these results serve as a baseline for subsequent results that do include simulated errors.

- **The retrieval error simulations of Fig. 3 look overly optimistic in comparison to the validation results of Wunch et al. (2017).**

Wunch et al. (2017) compare OCO-2 $XCO_2$ retrievals against $XCO_2$ observations at TCCON sites (the Total Column Observing Network). They report an average site bias of 0.22 ppm for comparisons between land nadir retrievals and TCCON sites. They also report an average root mean squared error of 1.31 ppm for the land nadir and TCCON comparisons (Table 3 in Wunch et al. 2017).

The errors in Fig. 3c-3f do appear slightly smaller than the numbers reported above. However, the errors in Fig. 3 are the mean of individual sounding errors in February and July, respectively – meaned within each PCTM grid box for an entire month. Hence, the errors displayed in this plot will be somewhat smaller than the errors on individual soundings (as reported in Wunch et al. 2017). By contrast, Fig. 1 shows the standard deviation of the estimated retrieval errors (instead of the mean as in Fig. 3). These standard deviations are larger than the mean and broadly consistent with the errors estimated by Wunch et al. (2017).

In the revised manuscript, we will clarify that the errors displayed in Fig. 3 are monthly means. Furthermore, we will compare and contrast the estimated errors with those estimated in Wunch et al. (2017) and in Worden et al. (2017).

- **Section 3.2 looks for flux patterns in $XCO_2$. Most top-down studies from OCO-2 would use a Bayesian approach where flux-error patterns are looked**

**for. This is more challenging because the signal is even smaller (while the paragraph in-between p. 5 and p. 6 suggests that the two approaches are rather equivalent with respect to the measurement information content). One should therefore discuss this limitation and further tone down the conclusions of the paper.**

The reviewer makes a great point, and we will add a discussion of this point to Sects. 2.2 (pgs. 5-6) and 3.2. The approach used here searches for flux patterns as they manifest in $XCO_2$. Phrased differently, the approach examines $s$ as seen through the OCO-2 observations, where $s$ are the fluxes. A Bayesian approach, by contrast, estimates $s - s_p$, where $s_p$ is the prior flux estimate. This residual flux ($s - s_p$) is presumably smaller than the total flux ($s$). As a result, inversions essentially estimate a smaller flux signal than the flux signal examined in this study.

The reviewer's argument could therefore imply more pessimistic results than presented in the current manuscript – that the $CO_2$ flux constraint is weaker than reported in the present study. This issue, however, may also be more nuanced. If the prior estimate is poor, the residual flux ($s - s_p$) will be large. These large flux patterns should be relatively easy to detect using $XCO_2$ observations, but the inversion will need to rely heavily on the $XCO_2$ observations (and not on the prior) to make a robust posterior estimate. By contrast, if the prior estimate is very accurate, the residual flux ($s - s_p$) will be small. The inversion will need to estimate a small flux signal, a signal that may be difficult to parse using $XCO_2$ observations. However, the posterior flux estimate will still be relatively robust due to the accurate prior.

Furthermore, this issue is specific to the setup of each individual inverse model. For example, existing $CO_2$ inversions use a wide variety of prior flux models. Mueller et al. (2008) use a non-informative prior (i.e., a flat prior) that contributes little information on the fluxes. In addition, many geostatistical inverse modeling

studies use environmental driver data in place of a tradition prior flux estimate (e.g., Gourdji et al. 2008; 2012). These studies choose environmental driver data using a model selection approach in a manner that is somewhat akin to the current ACPD manuscript. In the present study, we instead try to examine more fundamental questions about the robustness of the flux constraint, questions that are independent of subjective choices specific to each inverse model setup.

**References**

Crowell, S., et al. (2017, October). The OCO-2 Level 4 Flux Product. Poster presented at the OCO-2 Science Team Meeting, Boulder, Colorado.

Ganesan, A. L., Manning, A. J., Grant, A., Young, D., Oram, D. E., Sturges, W. T., Moncrieff, J. B., O'Doherty, S. (2015). Quantifying methane and nitrous oxide emissions from the UK and Ireland using a national-scale monitoring network. Atmospheric Chemistry and Physics, 15(11), 6393–6406, doi:10.5194/acp-15-6393-2015.

Gourdji, S. M., Mueller, K. L., Schaefer, K., Michalak, A. M. (2008). Global monthly averaged $CO_2$ fluxes recovered using a geostatistical inverse modeling approach: 2. Results including auxiliary environmental data. Journal of Geophysical Research: Atmospheres, 113(D21), doi:10.1029/2007JD009733.

Gourdji, S. M., Mueller, K. L., Yadav, V., Huntzinger, D. N., Andrews, A. E., Trudeau, M., Petron, G., Nehrkorn, T., Eluszkiewicz, J., Henderson, J., Wen, D., Lin, J., Fischer, M., Sweeney, C., Michalak, A. M. (2012). North American $CO_2$ exchange: intercomparison of modeled estimates with results from a fine-scale atmospheric inversion. Biogeosciences, 9(1), 457–475, doi:10.5194/bg-9-457-2012.

Liu, J., Bowman, K. W., Lee, M., Henze, D. K., Bousserez, N., Brix, H., Collatz, G. J., Menemenlis, D., Ott, L., Pawson, S., Jones, D., Nassar, R. (2014). Carbon monitoring system flux estimation and attribution: impact of ACOS-GOSAT $XCO_2$ sampling on the inference of terrestrial biospheric sources and sinks. Tellus B: Chemical and Physical

[Figure]

Meteorology, 66(1), 22486, doi:10.3402/tellusb.v66.22486.

Liu, J., Bowman, K. W., Schimel, D. S., Parazoo, N. C., Jiang, Z., Lee, M., Bloom, A. A., Wunch, D., Frankenberg, C., Sun, Y., O, C. W., Gurney, K. R., Menemenlis, D., Gierach, M., Crisp, D., Eldering, A. (2017). Contrasting carbon cycle responses of the tropical continents to the 2015–2016 El Niño. Science, 358(6360), doi:10.1126/science.aam5690.

Mueller, K. L., Gourdji, S. M., Michalak, A. M. (2008). Global monthly averaged $CO_2$ fluxes recovered using a geostatistical inverse modeling approach: 1. Results using atmospheric measurements. Journal of Geophysical Research: Atmospheres, 113(D21), doi:10.1029/2007JD009734.

Olsen, S. C. Randerson, J. T. (2004). Differences between surface and column atmospheric $CO_2$ and implications for carbon cycle research. Journal of Geophysical Research: Atmospheres, 109(D2), doi:10.1029/2003JD003968.

Rayner, P. J. O'Brien, D. M. (2001). The utility of remotely sensed $CO_2$ concentration data in surface source inversions. Geophysical Research Letters, 28(1), 175–178, doi:10.1029/2000GL011912.

Rigby, M., Manning, A. J., Prinn, R. G. (2011). Inversion of long-lived trace gas emissions using combined Eulerian and Lagrangian chemical transport models. Atmospheric Chemistry and Physics, 11(18), 9887–9898, doi:10.5194/acp-11-9887-2011.

Worden, J. R., Doran, G., Kulawik, S., Eldering, A., Crisp, D., Frankenberg, C., O'Dell, C., Bowman, K. (2017). Evaluation and attribution of OCO-2 $XCO_2$ uncertainties. Atmospheric Measurement Techniques, 10(7), 2759–2771, doi:10.5194/amt-10-2759-2017.

Wunch, D., Wennberg, P. O., Osterman, G., Fisher, B., Naylor, B., Roehl, C. M., O'Dell, C., Mandrake, L., Viatte, C., Kiel, M., Griffith, D. W. T., Deutscher, N. M., Velazco, V. A., Notholt, J., Warneke, T., Petri, C., De Maziere, M., Sha, M. K., Sussmann, R.,

Rettinger, M., Pollard, D., Robinson, J., Morino, I., Uchino, O., Hase, F., Blumenstock, T., Feist, D. G., Arnold, S. G., Strong, K., Mendonca, J., Kivi, R., Heikkinen, P., Iraci, L., Podolske, J., Hillyard, P. W., Kawakami, S., Dubey, M. K., Parker, H. A., Sepulveda, E., Garc, O. E., Te, Y., Jeseck, P., Gunson, M. R., Crisp, D., Eldering, A. (2017). Comparisons of the Orbiting Carbon Observatory-2 (OCO-2) $X_{CO_2}$ measurements with TCCON. Atmospheric Measurement Techniques, 10(6), 2209–2238, doi:10.5194/amt-10-2209-2017.

---

## Referee Comment (RC2) · Anonymous Referee #2 · 14 Nov 2017

This study presents a method to quantify the information that is provided by the greenhouse gas observing satellite OCO-2 on surface fluxes of CO2, as derived from the data using inverse modelling techniques. The approach is interesting because it provides an alternative to methods that have been used in the past for quantifying satellite instrument performance. Whereas those methods used uncertainty reduction to measure performance, this method quantifies the number of independent pieces of information provided by the data. The application to OCO-2 suggests that only a few independent pieces of information can be extracted from the data, mainly because of the size of the errors involved. In my opinion, this manuscript is suitable for publication in ACP provided that the following comments – mostly requests for further clarification

[Figure]

– are adequately addressed.

GENERAL COMMENTS

In Figure 3 the single sounding error of OCO-2 is compared to the signal from uncertainties in biospheric CO2 fluxes. The question is if this comparison makes much sense, since the error budget of OCO-2 has a large random component. The impact of biospheric flux uncertainties is more coherent in space and time, i.e. has very different statistics. Because of this the signal/noise ratio could look very different after space-time averaging of the data.

It is not clear to me what fraction of the flux uncertainty space is spanned by the flux patterns that are used in the regression. Probably many of the patterns are not independent, in which case it is not a surprise that many are not selected. This probably goes back to the question whether the range of estimates of the underlying models provides a fair estimate of the overall uncertainty. This is not easy to prove, but with only a single ocean pattern and a single anthropogenic emission pattern it seems conceivable that the uncertainty space is underestimated (by the way, how about uncertainties in land-use change?). Some discussion is needed of how such factors may influence the results, and what the implication could be for the estimated OCO-2 performance.

SPECIFIC COMMENTS

page 1, line 23: 'unlike previous missions' .. but this was the case also for GOSAT and SCIAMACHY.

page 2, line 12: references are needed to the recent special issue on OCO-2 in Science.

page 3, line 17-20: unless 'region' is defined more quantitatively these sentences are too vague.

page 3, line 19-23: Explain the motivation for this second approach? Is one considered to be more realistic than the other?

page 5, line 9: the constant fluxes need to be defined more quantitatively. What did you use? The same flux for each region and month? Are they estimated per region? Does it mean that the regressed flux patterns have zero mean? If so please mention.

page 7, line 25: should we conclude that OCO-2's glint mode retrievals do not provide significant independent information?

page 8, line 18: I would argue that the ocean is too strongly constraint by allowing only a single pattern to be adjusted in the regression. If more degrees of freedom would be assigned to the ocean, wouldn't that influence OCO-2's flux resolving performance over land?

page 8, line 21: this means that the biospheric flux patterns are specified per region and month, or?

page 8, line 27: 'stringent' in what sense? (I'd say they are rather less well contraint)

page 8, line 31: Would this goal be achieved if the 7 biomes could be resolved by OCO-2? Some quantitative information on how to relate surface and satellite measurements is needed here.

page 9, line 26-32: Should the reader conclude from this that we don't know whether the signal/noise analysis in figure 3 means anything?

page 10, line 16: 'scales smaller than hemispheric in about half of the cases'. How can you infer information about hemispheres from a split between Tropics and Extra Tropics? The way I look at it only a single pattern is selected in 3 out of 4 seasons. Is that sufficient to resolve two pieces of information? The text suggests that OCO-2 does better than 2 ...

page 10, line 18: 'we choose flux patterns ...' does this mean 1 or more?

page 10, line 32: Why is n* going down with the number of regions? Wouldn't you expect the residuals to become more random when fitting more regions? Shouldn't

that make V more diagonal?

page 11, line 31: Or underestimate noise? Is there a factor in the synthetic experiments that accounts for retrieval noise?

page 11, line 33: It doesn't really become clear what is mean by this "salient role". Can this be seen in the presented results?

page 12, line 19: Does the relative role of transport and measurement uncertainty follow from the results of this study, or is this just speculation? It seems to me that the study should provide information on this.

page S4, line 141: 'Consistency check'. What potential inconsistency is checked? Do you mean sensitivity or robustness check?

TECHNICAL CORRECTIONS

page 2, line 7: 'the the'

---

## Author Response (AR1)

We would like to thank the reviewer for ideas and suggestions for the manuscript. This feedback will be very helpful for updating and improving the manuscript. Below, we have included both the reviewer's suggestions (in bold) along with the associated changes we plan to make.

- **I found it particularly difficult to follow the logic of the paper and to evaluate the soundness of the approach. As a preliminary step for publication, the authors should seriously invest in making their study accessible to the**

[Figure]

**broad audience of ACP.**

This is very helpful feedback for framing the paper and describing the components of the methodology. We plan to re-frame the paper logic in several ways to make it more accessible to a broad audience. First, we plan to expand the overall description of the paper and description of the general approach at the end of the introduction (pg. 2, line 28 to pg. 3, line 21). We will describe the paper narrative in non-technical terms to give the reader an intuitive, high-level overview of the paper logic and flow. This description would provide better intuition for a wide audience of readers, especially those readers who may skip over the more technical information in the methodology (sect. 2).

Second, we will simplify the methods section (sect. 2) so that it is accessible to a broad audience. For example, this section contains seven equations. We will move several of these equations to the SI (e.g., Eq. 4-7) and instead expand the non-technical portions of the description. In this way, the paper will still include all of the technical detail for readers who want it, but the description in the main paper will be accessible to a broader audience.

Third, we will provide more references to existing studies that use similar approaches. Readers who are interested in more details on the methodology could gain greater context using these references. We will make this change throughout the manuscript and particularly from pg. 6, line 10 to pg. 7, line 20.

- **The paper concludes to a limited utility of OCO-2 retrievals for flux estimation with current retrieval algorithms and transport model. This may be correct, but is orthogonal to the claim made by Liu et al. (2017). The disagreement should be clearly stated.**

We will discuss this difference in the revised version of the manuscript. Liu et al. (2017) was published after this ACPD manuscript, so it is only now possible to make this comparison. Liu et al. use an atmospheric inversion to estimate

$CO_2$ fluxes for different tropical regions of the globe. They estimate uncertainties in their regional budget estimates, and those uncertainties are generally smaller than implied by the current ACPD manuscript.

Liu et al. (2017) use a 4DVAR approach and estimate the posterior uncertainties using a small number of Monte Carlo simulations. However, these uncertainty estimates are likely to be underestimates – due to compromises required to make the inversion computationally tractable. For example, most satellite-based inversions like Liu et al. (2017) do not fully account for error correlations or biases in the observations and atmospheric model; these studies typically use a diagonal error covariance matrix. Furthermore, Liu et al. (2017) and other studies use a small number of Monte Carlo simulations to estimate the errors (e.g., 60 simulations in Liu et al. (2014)). By contrast, Ribgy et al. (2011) and Ganesan et al. (2014) argue that 100,000 and 25,000 realizations are necessary to robustly estimate uncertainties for their particular inverse modeling problems. Note that it is not always possible to generate large numbers of realizations or fully account for error correlations in current satellite-based inverse models due to computational constraints. In the ACPD manuscript, we do not use a 4DVAR inverse model for this reason.

Consistent with this interpretation, results from the OCO-2 flux team ongoing intercomparison study indicate much larger uncertainties in estimated fluxes. The results are broadly consistent with those presented in the current ACPD manuscript (e.g., Crowell et al. 2017); preliminary results indicate that OCO-2 observations currently provide robust constraints for hemispheric regions but provide weaker constraints for individual continents or subcontinents. More specifically, recent flux team comparisons include $CO_2$ flux estimates from about eight different inverse modeling groups, and the level of disagreement among these estimates provides a measure of uncertainty in current top-down flux estimates that use the same version of the OCO-2 retrievals as applied in the current work

and in Liu et al. (2017). These estimates (using nadir observations) often show relatively good agreement for total hemispheric terrestrial $CO_2$ budgets, with the disagreement among inverse modeling estimates being smaller than the total $CO_2$ budget for a given hemisphere. The opposite is often true of $CO_2$ budgets estimated for smaller regions (e.g., Sub-Saharan Africa or Tropical Asia), with the disagreement among inverse modeling estimates usually being larger than the total budget. This ongoing work is consistent with the interpretation in the current manuscript.

- **Section 3.1 and the first part of Section 3.3 reinvent the wheel. See, e.g., Olsen and Randerson (2004) and Worden et al. (2017). Similarly, l. 23-28 are just an adaptation of an old argument (Rayner and O'Brien, 2001).**

The studies mentioned above investigate several requirements for constraining carbon budgets with satellite observations. Rayner and O'Brien (2001) explore the measurement precision required for space-based constraints on surface $CO_2$ fluxes. Olsen and Randerson (2004) model $XCO_2$ column enhancements across the globe due to surface $CO_2$ fluxes and compare them with surface enhancements. Lastly, Worden et al. (2017) estimate the errors in OCO-2 $XCO_2$ observations.

As the reviewer points out, the concepts used in Sects. 3.1 and 3.3 are, in part, built on these earlier approaches. However, the purpose of this section is not to develop new concepts. Rather, we build on existing concepts to assess real OCO-2 data. Rayner and O'Brien (2001) and Olsen and Randerson (2004), by contrast, did not have any real $XCO_2$ observations at their disposal, only simulations of possible future observations. Furthermore, we feel that these sections provide useful context and improve the manuscript narrative. Much of the manuscript presents the results of statistical experiments. These experiments use, as inputs, $XCO_2$ observations from OCO-2 and estimates of atmospheric transport and satellite retrieval errors. Sect. 3.1 provides visualizations of those

inputs.

In the revised manuscript, we will cite the studies listed above, clarify that we use concepts from the studies, and explain that we apply those concepts to real observations from OCO-2. We will also compare the retrieval errors in Sect. 3.1 against those in Worden et al. (2017). Lastly, we will shorten the first part of Sect. 3.3. That section presents the synthetic study results with no errors; these results serve as a baseline for subsequent results that do include simulated errors.

- **The retrieval error simulations of Fig. 3 look overly optimistic in comparison to the validation results of Wunch et al. (2017).**

Wunch et al. (2017) compare OCO-2 $XCO_2$ retrievals against $XCO_2$ observations at TCCON sites (the Total Column Observing Network). They report an average site bias of 0.22 ppm for comparisons between land nadir retrievals and TCCON sites. They also report an average root mean squared error of 1.31 ppm for the land nadir and TCCON comparisons (Table 3 in Wunch et al. 2017).

The errors in Fig. 3c-3f do appear slightly smaller than the numbers reported above. However, the errors in Fig. 3 are the mean of individual sounding errors in February and July, respectively – meaned within each PCTM grid box for an entire month. Hence, the errors displayed in this plot will be somewhat smaller than the errors on individual soundings (as reported in Wunch et al. 2017). By contrast, Fig. 1 shows the standard deviation of the estimated retrieval errors (instead of the mean as in Fig. 3). These standard deviations are larger than the mean and broadly consistent with the errors estimated by Wunch et al. (2017).

In the revised manuscript, we will clarify that the errors displayed in Fig. 3 are monthly means. Furthermore, we will compare and contrast the estimated errors with those estimated in Wunch et al. (2017) and in Worden et al. (2017).

- **Section 3.2 looks for flux patterns in $XCO_2$. Most top-down studies from OCO-2 would use a Bayesian approach where flux-error patterns are looked**

[Figure]

**for. This is more challenging because the signal is even smaller (while the paragraph in-between p. 5 and p. 6 suggests that the two approaches are rather equivalent with respect to the measurement information content). One should therefore discuss this limitation and further tone down the conclusions of the paper.**

The reviewer makes a great point, and we will add a discussion of this point to Sects. 2.2 (pgs. 5-6) and 3.2. The approach used here searches for flux patterns as they manifest in $XCO_2$. Phrased differently, the approach examines $s$ as seen through the OCO-2 observations, where $s$ are the fluxes. A Bayesian approach, by contrast, estimates $s - s_p$, where $s_p$ is the prior flux estimate. This residual flux $(s - s_p)$ is presumably smaller than the total flux $(s)$. As a result, inversions essentially estimate a smaller flux signal than the flux signal examined in this study.

The reviewer's argument could therefore imply more pessimistic results than presented in the current manuscript – that the $CO_2$ flux constraint is weaker than reported in the present study. This issue, however, may also be more nuanced. If the prior estimate is poor, the residual flux $(s - s_p)$ will be large. These large flux patterns should be relatively easy to detect using $XCO_2$ observations, but the inversion will need to rely heavily on the $XCO_2$ observations (and not on the prior) to make a robust posterior estimate. By contrast, if the prior estimate is very accurate, the residual flux $(s - s_p)$ will be small. The inversion will need to estimate a small flux signal, a signal that may be difficult to parse using $XCO_2$ observations. However, the posterior flux estimate will still be relatively robust due to the accurate prior.

Furthermore, this issue is specific to the setup of each individual inverse model. For example, existing $CO_2$ inversions use a wide variety of prior flux models. Mueller et al. (2008) use a non-informative prior (i.e., a flat prior) that contributes little information on the fluxes. In addition, many geostatistical inverse modeling

studies use environmental driver data in place of a tradition prior flux estimate (e.g., Gourdji et al. 2008; 2012). These studies choose environmental driver data using a model selection approach in a manner that is somewhat akin to the current ACPD manuscript. In the present study, we instead try to examine more fundamental questions about the robustness of the flux constraint, questions that are independent of subjective choices specific to each inverse model setup.

[Figure]

**random component. The impact of biospheric flux uncertainties is more coherent in space and time, i.e. has very different statistics. Because of this the signal/noise ratio could look very different after space-time averaging of the data.**

Figure 3 in the current ACPD manuscript shows the mean of all soundings in each PCTM model grid box for February and July, respectively. Reviewer 1 brought up this question as well, and we will clarify this point in the revised manuscript.

As the reviewer points out, the signal-to-noise ratio in Fig. 3 will vary depending on space-time averaging. With that said, many inverse modeling studies report monthly $CO_2$ flux totals, so the monthly averaging in Fig. 3 is particularly pertinent. Furthermore, the uncertainties in top-down $CO_2$ flux estimates change when averaged to aggregate space-times scales, so this issue is also a consideration in inverse modeling, not just the analysis in Fig. 3.

We will revise the discussion of Fig. 3 in several ways to account for the reviewer's suggestion. First, we will explain that the signal-to-noise ratio varies depending upon the space and time scales considered, and we will explain why this monthly scale is a particularly useful time period to examine. Second, we will emphasize that this signal-to-noise ratio provides a useful intuition or feel for the data, but we will point out that top-down inverse models leverage the signal in much more sophisticated ways. The limitations of this signal-to-noise comparison thus motivate subsequent analyses in the manuscript.

- **It is not clear to me what fraction of the flux uncertainty space is spanned by the flux patterns that are used in the regression. Probably many of the patterns are not independent, in which case it is not a surprise that many are not selected. This probably goes back to the question whether the range of estimates of the underlying models provides a fair estimate of the overall uncertainty. This is not easy to prove, but with only a single ocean pattern and a single anthropogenic emission pattern it seems conceivable that the**

[Figure]

**uncertainty space is underestimated (by the way, how about uncertainties in land-use change?). Some discussion is needed of how such factors may influence the results, and what the implication could be for the estimated OCO-2 performance.**

This factor can influence the results, and we will add a discussion of this point to the manuscript. We explore this possibility in the synthetic data experiments (Fig. 5b in the ACPD manuscript). In that experiment, we create synthetic $XCO_2$ observations using the SiBCASA flux model and an atmospheric transport model. We then run model selection, but we do not include SiBCASA as a possible predictor variable in the regression. In other words, model selection can include several different terrestrial biosphere models (TBMs) in the regression, but it cannot include the TBM that was used to generate the synthetic data in the first place. Fig. 5b in the current ACPD manuscript shows the result. Model selection does not select patterns in every region and every month, but it still selects flux patterns for most regions and months.

This issue also affects Bayesian inverse models. These inversions use a prior flux estimate as an initial guess for the fluxes. If the prior flux estimate is inaccurate, the prior error covariance matrix will have large variances/covariances, and the posterior uncertainties will likely be large. If the prior flux estimate is skilled, the prior error covariance matrix will have small variances/covariances, and the posterior uncertainties will be smaller. In other words, the availability and skill of prior flux models (i.e., TBMs) affects the robustness and uncertainty of the inverse modeling estimate.

• **SPECIFIC COMMENTS**

• **page 1, line 23: 'unlike previous missions' .. but this was the case also for GOSAT and SCIAMACHY.**

We will change the text accordingly. In the revised text, we will remove the

phrase "Unlike previous missions" and briefly explain the similarities and differences among OCO-2, GOSAT, and SCIAMACHY.

- **page 2, line 12: references are needed to the recent special issue on OCO-2 in Science.**

We will rewrite this paragraph and discuss studies from the new *Science* special issue. This special issue was published after the present ACPD manuscript, and it is now possible to reference these papers in the manuscript.

- **page 3, line 17-20: unless 'region' is defined more quantitatively these sentences are too vague.**

We will revise these sentences accordingly. In response to feedback from reviewer 1, we plan to re-write the second half of Sect. 1 to describe the overall objectives and approach in a way that is more accessible to a broad audience. To that end, we will more concisely define the word "region".

- **page 3, line 19-23: Explain the motivation for this second approach? Is one considered to be more realistic than the other?**

We will clarify the text in this paragraph. We do not consider one approach to be more accurate than another per se. Rather, it is challenging to estimate realistic retrieval errors because these errors are unknown (except possibly at TCCON sites). We asked several colleagues for advice on how to estimate these errors, and different colleagues recommended different approaches that produce different retrieval error estimates. As a result, we decided to use two different retrieval error estimates – to ensure that the results were not contingent upon the specific method used.

- **page 5, line 9: the constant fluxes need to be defined more quantitatively. What did you use? The same flux for each region and month? Are they**

[Figure]

**estimated per region? Does it mean that the regressed flux patterns have zero mean? If so please mention.**

We will clarify this topic in the manuscript, and we will define these constant terms more quantitatively.

The constant flux is estimated for each region and each month. This constant flux is included as a predictor variable in the regression, and the regression framework scales the magnitude of the constant flux in each region and month to match the observations.

Equations 1 and 2 in the manuscript describe the overall regression and illustrate these relationships quantitatively:

$Y = h(\mathbf{X})$

$z = \mathbf{Y}\boldsymbol{\beta} + \boldsymbol{\epsilon}$

where $\mathbf{X}$ are the predictor variables in the regression, $h()$ is the atmospheric transport model, $z$ are the observations, $\boldsymbol{\beta}$ are the coefficients estimated by the regression, and $\epsilon$ are the regression residuals. In this setup, the constant flux terms are individual columns in $\mathbf{X}$. Each column has a value of one in a given region and month and has values of zero elsewhere. Phrased differently, these constant flux terms are analogous to the y-intercept terms in the regression. Also of note, the regression residuals $\epsilon$ have a mean of zero, but the regressed flux patterns will not have a zero mean.

We will make several changes to clarify this topic in the manuscript. We will move Eqs. 1-2 earlier in Sect. 2.2 and describe these equations alongside the description of the constant or intercept terms. In response to reviewer 1, we will move several equations to the SI and simplify the description in Sect. 2.2. Instead, we will dedicate more description to Eqs. 1-2 and will explain how the different predictor variables (including the constant or intercept terms) fit into these equations.

[Figure]

- **page 7, line 25: should we conclude that OCO-2's glint mode retrievals do not provide significant independent information?**

  We will provide additional discussion of this point in the manuscript. The statement above may be too bold to make in the manuscript, especially in context of the reviewer's next suggestion below. Furthermore, the OCO-2 nadir and glint observations have different biases in the version 7 OCO-2 data product (the product used in this manuscript), and these differing biases make it difficult to use both types of observations in the same analysis. For example, there is a step change in the $XCO_2$ observations at the coastline in some locations (e.g., in parts of Africa). In these cases, the nadir mode observations may be sensitive to flux patterns, and the glint mode observations might be sensitive to flux patterns. However, an inverse model that uses both observation types together might produce unrealistic flux patterns due to the step change in $XCO_2$ at the coastline.

- **page 8, line 18: I would argue that the ocean is too strongly constraint by allowing only a single pattern to be adjusted in the regression. If more degrees of freedom would be assigned to the ocean, wouldn't that influence OCO-2's flux resolving performance over land?**

  We will add this caveat to the manuscript. If there are large, unresolved $CO_2$ fluxes from the ocean, it could influence top-down inferences of terrestrial biospheric fluxes. With that said, ocean fluxes on sub-daily time scales are much smaller than terrestrial fluxes, and the spatial patterns in these fluxes are much more diffuse than in most terrestrial regions. As a result, small errors in the distribution of marine $CO_2$ fluxes should not dramatically change the detectability of terrestrial fluxes.

- **page 8, line 21: this means that the biospheric flux patterns are specified per region and month, or?**

  This is correct. We tag $CO_2$ fluxes from each region and each month in the

[Figure]

PCTM atmospheric transport model. In other words, we incorporate flux patterns into PCTM at the PCTM model resolution; the model ingests $CO_2$ fluxes at a 1° latitude by 1.25° longitude spatial resolution and 3-hourly time resolution (Sect. 2.4 of the current ACPD manuscript). We then run the PCTM model once for each region and each month of interest. For each of these PCTM runs, we input flux patterns for the region and month of interest and zero out $CO_2$ fluxes for other regions and months. We will clarify that point in the associated paragraph of the revised manuscript.

- **page 8, line 27: 'stringent' in what sense? (I'd say they are rather less well contraint)**

We agree that "stringent" is not be the best or most descriptive word here. We will replace the word "stringent" with the following phrase: "This case is more demanding of the observations than the two and four region cases; it is more difficult to obtain a robust constraint for seven regions than for two or four global regions."

- **page 8, line 31: Would this goal be achieved if the 7 biomes could be resolved by OCO- 2? Some quantitative information on how to relate surface and satellite measurements is needed here.**

We will remove this sentence from the revised manuscript. Fang et al. (2014) examine $CO_2$ fluxes for North American biomes while the present ACPD manuscript focuses on global biomes. Hence, the two studies are not equivalent.

- **page 9, line 26-32: Should the reader conclude from this that we don't know whether the signal/noise analysis in figure 3 means anything?**

We think that interpretation would be too bold. We feel that the signal/noise analysis provides useful context; it is useful to show the reader what the biospheric $XCO_2$ signal looks like, how it varies across the globe, and how it varies by month.

[Figure]

The results in Sect. 3.2 and 3.3 are based on a statistical model, and we wanted to provide an intuitive illustration of the signal and noise before presenting statistical results that use those inputs.

- **page 10, line 16: 'scales smaller than hemispheric in about half of the cases'. How can you infer information about hemispheres from a split between Tropics and Extra Tropics? The way I look at it only a single pattern is selected in 3 out of 4 seasons. Is that sufficient to resolve two pieces of information? The text suggests that OCO-2 does better than 2 ...**

The reviewer makes a good point. A pattern is selected in approximately half of the regions and months. However, in three of the four seasons, not a single pattern is selected for one of the two hemispheres. We will add this description to the text to better represent the results.

- **page 10, line 18: 'we choose flux patterns ...' does this mean 1 or more?**

This statement is correct. We will revise this paragraph accordingly by changing "flux patterns" to "at least one flux pattern."

- **page 10, line 32: Why is n* going down with the number of regions? Wouldn't you expect the residuals to become more random when fitting more regions? Shouldn't that make V more diagonal?**

There are more unexplained flux patterns in the 7-region case – because model selection selects fewer variables than in the two or four region cases. As a result, the regression residuals have large covariances, and $\mathbf{V}$ is less diagonal. The variable $n^*$ is smaller as a result.

A brief overview of the regression helps elucidate why this is the case. The regression is iterative. We make an initial guess for $n^*$, run the regression with model selection, adjust $n^*$, and rerun the regression with model selection. We continue iterating until $n^*$ and the regression converge – until they stop changing

from one iteration to the next. As a result, the estimate for $n^*$ depends on which variables are included in the regression. We select a relatively small number of variables in the 7-region case, so there are many unexplained patterns in the residuals. The estimate for $n^*$ is smaller as a result.

- **page 11, line 31: Or underestimate noise? Is there a factor in the synthetic experiments that accounts for retrieval noise?**

  The reviewer makes a great point; the estimated retrieval errors could overestimate the covariances but underestimate the variances (i.e., white noise). We will add a sentence to the paragraph explaining this point.

- **page 11, line 33: It doesn't really become clear what is mean by this "salient role". Can this be seen in the presented results?**

  This statement references the synthetic data experiments in Fig. 5. In the revised manuscript, we will specifically reference the synthetic data experiments and Fig. 5.

- **page 12, line 19: Does the relative role of transport and measurement uncertainty follow from the results of this study, or is this just speculation? It seems to me that the study should provide information on this.**

  We will clarify this result in the revised manuscript. We explore the relative roles of transport and measurement/retrieval uncertainty in the synthetic data experiments (e.g., Fig. 5 in the current ACPD manuscript). In the revised manuscript, we will make reference to the figure here and explicitly tie this statement back to the synthetic data experiments.

- **page S4, line 141: 'Consistency check'. What potential inconsistency is checked? Do you mean sensitivity or robustness check?**

  We agree that it is better to use the term "sensitivity" or "robustness" instead of "consistency." We will change the text accordingly.

[Figure]

- **TECHNICAL CORRECTIONS**

- **page 2, line 7: 'the the'**

  Thank you for pointing out this typo. We will correct it in the revised manuscript.

  ───────────────────────────

[revised manuscript text omitted]

The synthetic and real data simulations are based upon a multiple regression combined with model selection. We use this approach to evaluate whether spatial and temporal patterns in biospheric $CO_2$ fluxes help describe patterns in the OCO-2 observations. These flux patterns are first input into a global atmospheric transport model (the Parameterized Chemistry Transport Model or are then compared against OCO-2 observations. A positive result implies that we can detect these patterns using OCO-2 observations, and a negative result implies that we cannot. Several existing studies use model selection to gauge the detectability of $CO_2$ flux patterns (Shiga et al., 2014; Fang et al., 2014; ASCENDS Ad Hoc Science Definition Team, 2015). The term 'patterns'

here refers to flux patterns that manifest at the resolution of an atmospheric model, and section 2 describes this approach in more detail.

Overall, we divide the globe into different hemispheres and ecoregions and determine whether we can detect flux patterns within each region and each month. We begin the analysis with very large hemispheric regions and then decrease the size of those regions until we are no longer able to detect any patterns beyond a mean $CO_2$ flux. That limit or end point is the smallest scale at which OCO-2 observations currently provide a unique constraint on $CO_2$ budgets. OCO-2 observations must be sufficient to detect more than a mean flux across a region and month if future inverse modeling studies are to estimate biospheric $CO_2$ budgets at scales smaller than that region. Consequently, inverse modeling studies would generally be unable to obtain reliable information about the fluxes across smaller regions. This result bounds the type of information one can expect from the OCO-2 retrievals in their current stage of development.

**2  Methods**

**2.1  OCO-2 data**

This study utilizes $XCO_2$ observations from the OCO-2 satellite beginning with the first reported data (6 Sept. 2014) through the end of 2015. We use the level 2 lite product, version B7.1.01; the lite product only includes good quality retrievals, unlike the full OCO-2 level 2 product. We only include nadir and target mode retrievals in the analysis and exclude glint mode retrievals. Recent work indicates biases in the glint retrievals relative to nadir retrievals (e.g., Wunch et al., 2017a). The SI describes model selection results with glint mode retrievals included, and the results are similar to those in the main manuscript without glint mode data.

**2.2  Simulated model and retrieval errors**

We simulate different model and measurement errors and compare those errors against a modeled $XCO_2$ signal from biospheric fluxes 
[revised manuscript text omitted]

25

30   ~~We subsequently model $XCO_2$ using an atmospheric transport model (Sect. ??) and several different biospheric flux estimates and vegetation indices. We then incorporate these model outputs as predictor variables in a regression and use model selection to identify which patterns (if any) explain substantial variability in the OCO-2 observations. These patterns include four TBMs with contrasting spatial features from MsTMIP, the Multi-scale Synthesis and Terrestrial Model Intercomparison Project (Huntzinger et al., 2013; Fisher et al., 2016; ?). Section S1.2 describes MsTMIP and the TBMs in greater detail. We also include SIF (solar-induced fluorescence) from the Global Ozone Monitoring Experiment-2 (GOME-2, Joiner et al., 2013) as well as EVI (enhanced vegetation index) and NDVI (normalized difference vegetation index) from the Moderate-Resolution~~

Imaging Spectroradiometer (MODIS; e.g., Huete et al., 2002). Note that we directly input these vegetation indices into an atmospheric transport model as a surface 'flux.' The regression/model selection framework will adjust the magnitude of the transport model outputs to reproduce the OCO-2 observations, so the absolute magnitude of the vegetation indices is not important. Rather, we are interested in whether the patterns in these vegetation indices help reproduce patterns in the OCO-2 observations, potentially in combination with other indices or TBMs.

We offer up a relatively large number of flux models and vegetation indices as predictor variables, and at least some of these products are therefore expected to correlate with real world $CO_2$ fluxes. We should choose at least one of these variables using model selection if the OCO-2 observations are able to detect patterns in the surface fluxes. If we do not choose any additional outputs with model selection, it suggests that the observations are not sufficient to detect spatial and temporal patterns in the fluxes beyond a mean flux. We include a large number of candidate variables for a pragmatic reason. If model selection does not pick any variable in a region, it is unlikely that there was a shortage of reasonable $CO_2$ flux patterns available to choose from. Rather, that result more likely reflects the maturity of current OCO-2 observations and atmospheric modeling capabilities.

 Model selection provides a convenient way to evaluate the information content of OCO-2 observations in their current state of development. In theory, one could estimate $CO_2$ budgets in a Bayesian inverse model. The accuracy or uncertainty in those budgets would be indicative of the information content of the satellite observations. This approach, however, brings several challenges. First, a modeler must choose a prior flux estimate. This choice is often subjective but will impact the final or posterior uncertainty estimate (e.g., Chevallier et al., 2014). Second, a modeler must estimate several individual sources of uncertainty as inputs to the inverse model. These uncertainties often have a complex statistical structure that is difficult to characterize (e.g., Liu et al., 2011), and it is often challenging to account for all plausible sources of uncertainty. Third, inverse modeling with satellite observations can be computationally intensive – both in terms of the number of atmospheric model simulations required and the computational requirements of the statistical inverse model. Some studies have overcome the first of these two challenges by using an ensemble of atmospheric models and/or inversion systems (e.g., Chevallier et al., 2014; Houweling et al., 2015). The size of the ensemble spread is indicative of the information content of the observations, and the spread of the ensemble is usually larger than the uncertainty bounds estimated from any one inverse model. This type of study also typically requires extensive coordination among multiple research groups. Model selection, by contrast, By contrast, the regression framework used here provides a simpler metric to evaluate the information content of the observations.

The remainder of this sub-section describes the specific equations used for model selection . We quantitatively link the OCO-2 $XCO_2$ observations with model outputs using a multiple regression :

$$\mathbf{Y} \quad = h(\mathbf{X})$$

$$\mathbf{z} \quad = \mathbf{Y}\boldsymbol{\beta} + \boldsymbol{\epsilon}$$

$$\boldsymbol{\epsilon} \quad \sim \mathcal{N}(\mathbf{0}, \sigma^2 \mathbf{V})$$

In these equations, the vector $\mathbf{z}$ (dimensions $n \times 1$)represents the $XCO_2$ observations minus the model initial condition or spin-up (refer to the SI for more detail). The variable

**2.4 Implementation of the top-down experiments**

This section describes how the regression and model selection are implemented in the present study.

The regression begins with an intercept. The intercept is always included in the regression (in $\mathbf{X}$), and model selection can further add flux models to $\mathbf{X}$  to help reproduce the OCO-2 ~~retrievals, and the resulting matrix $\mathbf{Y}$ has dimensions $n \times p$. The variable $\epsilon$ is a $n \times 1$ vector of residuals. These residuals are assumed to follow a multivariate normal distribution with a mean of zero, a variance of $\sigma^2$, and a covariance structure given by $\mathbf{V}$ (dimensions $n \times n$). The vector of coefficients ($\boldsymbol{\beta}$, dimensions $p \times 1$) are estimated as part of the regression .~~

 observations ($\boldsymbol{z}$). This intercept is a constant column of ones in $\mathbf{X}$ In the particular setup here, we include a different intercept for each region of the globe and each month. In other words, the intercept consists of multiple columns – one column for each region and month of the study period. This intercept is equivalent to a spatially and temporally constant flux in each region and month. Additional atmospheric model outputs $h(\mathbf{X})$ will not be selected unless they explain substantially more variability in the OCO-2 observations than this intercept or constant flux. The intercept plays an important role in the regression; it ensures that the regression will always be unbiased when averaged across the globe. Model selection could produce non-intuitive results if there were no intercept. Furthermore, intercepts are standard in regression modeling, and

$$BIC = L + p \ln(n^*)$$

top-down, $CO_2$ studies that utilize model selection also include an intercept (e.g., Gourdji et al., 2008, 2012; Fang et al., 2014)

We subsequently run the regression with model selection three times to evaluate three different cases. In the first case, we divide the flux models into two large hemispheric regions, and we select different flux models for each hemispheric region and each month. The second and third experiments divide the fluxes into four continental regions and seven ecoregions, respectively (Fig. 2). The

$$L \qquad = n^* \ln(\sigma^2) + \frac{n^*}{n} RSS$$

$$RSS \quad = \frac{1}{\sigma^2} \boldsymbol{z}^T \boldsymbol{z} - \frac{1}{\sigma^2} \boldsymbol{z}^T \mathbf{Y} (\mathbf{Y}^T \mathbf{Y})^{-1} \mathbf{Y}^T \boldsymbol{z}$$

 last experiment is more challenging than the first.

5  Note that we consider the model selection results from 2014 part of an initial model spin-up period and only report the results from 2015.

**2.5  Synthetic data simulations**

We subsequently utilize synthetic data simulations in this study to analyze the effects of different model or retrieval errors on the detectability of biospheric $CO_2$ fluxes. In these simulations, we create synthetic $XCO_2$ observations using an atmospheric

10  transport model (Sect. 2.6) and the SiBCASA flux model. We then run model selection using these synthetic observations in place of real OCO-2 observations ($z$). These model selection runs have the same setup as the real data simulations, except that the observations are synthetic instead of real. Furthermore, we only analyze the seven ecoregion case in the synthetic data experiments. This case is more demanding of the observations than the two and four region cases; it is more difficult to obtain a robust constraint for seven regions than for two or four larger global regions. The seven ecoregion case is also an important goal

15  from a ecological perspective. For example, one might want to estimate how $CO_2$ fluxes differ in different tropical forests or in different temperate forests (e.~~Jones (2011) discusses this concept in the context of the BIC. Just because the satellite provides $n$ observations does not mean there are $n$ independent pieces of information. Accordingly, $n^*$ ensures that the model selection framework accurately assesses the amount of independent information in the observations . It accounts for the fact that there are often spatial and temporally coherent errors in the satellite observations or in the transport model. If all of the observations~~

20  g., on different continents or in different climate zones).

 These synthetic simulations help to isolate the effect of different errors on the detectability of biospheric $CO_2$

25

$$n^* = \frac{n}{1 + \left( \sum_{i=1}^{n} \sum_{j=1, j \neq i}^{n} V_{i,j} / n \right)}$$

30

**2.6**

[revised manuscript text omitted]

25  ~~same setup as the real data simulations, except that the observations ($z$) are synthetic instead of real. Furthermore, we only analyze the seven region case in the synthetic data experiments. This case is more stringent than the two and four region cases. The seven ecoregion case is also an important goal from a ecological perspective: one might want to estimate how $CO_2$ fluxes differ in different tropical forests or in different temperate forests (e.g., on different continents or in different climate zones). The in-situ monitoring network in North America, for example, is able to detect flux patterns within most North American~~

30

~~The first model selection runs are an idealized case with no simulated errors ($\epsilon \approx 0$). We then successively add simulated error to the synthetic observations and evaluate how the model selection results change as the errors increase (Fig. 1). We include three different types of errors: flux errors, transport errors, and observation or retrieval errors. Section 2.2 and the SI describe the simulated transport and retrieval errors. The flux errors further account for plausible inaccuracies in the predictor variables within $\mathbf{X}$. No TBM or vegetation index has a distribution that perfectly matches real-world $CO_2$ fluxes. These errors affect our ability to identify biospheric flux patterns using the OCO-2 observations and would affect inverse modeling studies; the better the prior flux estimate in an inverse model , the more accurate and reliable the posterior flux estimate. We therefore~~

5

[revised manuscript text omitted]
 sub-section describes the regression and model selection in greater detail. The regression used in this paper will quantitative link OCO-2 XCO$_2$ observations with atmospheric model outputs:

$$\mathbf{Y} = h(\mathbf{X}) \tag{S1}$$

$$\boldsymbol{z} = \mathbf{Y}\boldsymbol{\beta} + \boldsymbol{\epsilon} \tag{S2}$$

$$\boldsymbol{\epsilon} \sim \mathcal{N}(\mathbf{0}, \sigma^2 \mathbf{V}) \tag{S3}$$

These equations are an expanded form of the regression equations present in Sect. 2.3. The vector $\boldsymbol{z}$ (dimensions $n \times 1$) represents the XCO$_2$ observations minus the model initial condition or spin-up (Sect. S1.1). The variable $\mathbf{X}$ (dimensions $m \times p$) is a matrix of $p$ different flux models, and each column of $\mathbf{X}$ is a different flux model for a different region and month. The function $h()$ is an atmospheric model that transports the fluxes to the times and locations of the OCO-2 retrievals, and the resulting matrix $\mathbf{Y}$ has dimensions $n \times p$. Furthermore, the variable $\boldsymbol{\epsilon}$ is a $n \times 1$ vector of residuals. These residuals are assumed to follow a multivariate normal distribution with a mean of zero, a variance of $\sigma^2$, and a covariance structure given by $\mathbf{V}$ (dimensions $n \times n$). The vector of coefficients ($\boldsymbol{\beta}$, dimensions $p \times 1$) are estimated as part of the regression.

In this study, we choose a set of variables for $\mathbf{X}$ using model selection based on the Bayesian Information Criterion (BIC) (Schwarz, 1978). We calculate a BIC score for many different candidate models. Each candidate model has a different set of columns ($\mathbf{X}$) – different combinations of flux models in different geographic regions and in different months.

The best model has the lowest BIC score:

$$BIC = L + p\ln(n^*) \tag{S4}$$

where $L$ is the log likelihood of a particular candidate model ($\mathbf{X}$). The log-likelihood has the following form:

$$L = n^* \ln(\sigma^2) + \frac{n^*}{n} RSS \tag{S5}$$

$$RSS = \frac{1}{\sigma^2} \boldsymbol{z}^T \boldsymbol{z} - \frac{1}{\sigma^2} \boldsymbol{z}^T \mathbf{Y} (\mathbf{Y}^T \mathbf{Y})^{-1} \mathbf{Y}^T \boldsymbol{z} \tag{S6}$$

where RSS is the residual sum of squares and $\sigma^2$ is defined above in Eq. S3.

Both the BIC and log-likelihood equations (Eq. S4 and S6) incorporate $n^*$, the effective number of independent observations. Jones (2011) discusses this concept in the context of the BIC. Just because the satellite provides $n$ observations does not mean there are $n$ independent pieces of information. Accordingly, $n^*$ ensures that the model selection framework accurately assesses the amount of independent information in the observations. It accounts for the fact that there are often spatial and temporally coherent errors in the satellite observations or in the transport model. If all of the observations were independent (i.e., if $\mathbf{V}$ were diagonal), then $n^*$ would equal $n$. However, we de-weight both components of Eq. S4 as the covariances in $\mathbf{V}$ increase.

We could calculate $n^*$ directly using $\mathbf{V}^{-1}$ (Jones, 2011). In fact, several $CO_2$ model selection studies incorporate $\mathbf{V}^{-1}$ directly into the equation for RSS (e.g., Mueller et al., 2010; Gourdji et al., 2012; Shiga e We use 5,079,165 observations ($n$) in this study, so $\mathbf{V}$ has $5.08 \times 10^6$ rows and columns. As a result, the inverse of $\mathbf{V}$ is computationally intractable. We instead estimate $n^*$ using an approach based on Griffith (2005), an approach that does not require computing $\mathbf{V}^{-1}$ directly:

$$n^* = \frac{n}{1 + \left( \sum_{i=1}^n \sum_{j=1, j \neq i}^n V_{i,j}/n \right)} \tag{S7}$$

This equation calculates $n^*$ using individual elements of $\mathbf{V}$ and does not require inverting the full matrix; it is therefore far more computationally tractable. Subsequent paragraphs describe how we estimate the elements of $\mathbf{V}$.

The remainder of this section discusses how we characterize the variances ($\sigma^2$) and covariance structure ($\mathbf{V}$) of the  residuals ($\epsilon$) (Eq. S3). An estimate of the variance is required to calculate the residual sum of squares (RSS, Eq. S6), and an estimate of the covariance structure is necessary to calculate the effective number of independent observations ($n^*$, Eq. S7).

We first describe  the method for characterizing the covariance structure ($\mathbf{V}$). We estimate  the individual elements, $V_{i,j}$, in the vicinity of each  observation $i$  by fitting a local variogram model on the model-data residuals ($\epsilon$). The covariance structure likely differs in different locations and at different times (i.e., is non-stationary), and several existing studies fit variograms locally to account for this non-stationary structure  (e.g., Alkhaled et al., 2008; Hammerling et al., 2012; Tadić et al., 2017). We use a similar approach here. Specifically, 
[revised manuscript text omitted]

Alkhaled, A. A., Michalak, A. M., Kawa, S. R., Olsen, S. C., and Wang, J.-W.: A global evaluation of the regional spatial variability of column integrated $CO_2$ distributions, J. Geophys. Res. – Atmos., 113, doi:10.1029/2007JD009693, d20303, 2008.

Cihlar, J., Caramori, P. H., Scnuepp, P. H., Desjardins, R. L., and MacPherson, J. I.: Relationship between satellite-derived vegetation indices and aircraft-based $CO_2$ measurements, J. Geophys. Res. – Atmos., 97, 18 515–18 521, doi:10.1029/92JD00655, 1992.

Didan, K.: MYD13C1 MODIS/Aqua Vegetation Indices 16-Day L3 Global 0.05 Deg CMG V006, Tech. rep., NASA EOSDIS Land Processes DAAC, doi:10.5067/MODIS/MYD13C1. 006, 2015a.

Didan, K.: MOD13C1 MODIS/Terra Vegetation Indices 16-Day L3 Global 0.05 Deg CMG V006, Tech. rep., NASA EOSDIS Land Processes DAAC, doi:10.5067/MODIS/MOD13C1. 006, 2015b.

Gourdji, S. M., Mueller, K. L., Yadav, V., Huntzinger, D. N., Andrews, A. E., Trudeau, M., Petron, G., Nehrkorn, T., Eluszkiewicz, J., Henderson, J., Wen, D., Lin, J., Fischer, M., Sweeney, C., and Michalak, A. M.: North American $CO_2$ exchange: inter-comparison of modeled estimates with results from a fine-scale atmospheric inversion, Biogeosciences, 9, 457–475, doi:10.5194/bg-9-457-2012, 2012.

Griffith, D. A.: Effective geographic sample size in the presence of spatial autocorrelation, Ann. Assoc. Am. Geogr., 95, 740–760, doi:10.1111/j.1467-8306.2005.00484.x, 2005.

[revised manuscript text omitted]

---

## Author Response (AR2)

**Reply to reviewer feedback**

We thank the reviewer for suggestions on the article. The reviewer comments are in bold below with replies in regular text.

**The authors have improved the presentation of the paper much and have addressed previous issues well. However, there are a few minor issues left that should be corrected before publication:**
**p. 2, l. 12: Please remove "like those" since the statement is not valid for Tansat.**

We have replaced "like those" with "than".

**p. 4, l. 34: Please remove "unique" since other types of constraints (i.e. not from XCO2) are not discussed here.**

We have removed the word accordingly.

**p. 6, "This choice is often subjective but will impact the final or posterior uncertainty estimate (e.g., Chevallier et al., 2014)." This argument is expressed as one of the challenges, but is not: getting a prior flux estimate is trivial and the posterior errors do not depend on the prior flux estimate per se (in a usual linear Gaussian framework). The quoted reference does not seem to support the argument either and would be better placed in the other success/challenge list l. 2, l. 29.**

We have replaced the reference to Chevallier et al. (2014) with Peylin et al. (2013). This reference better supports the argument in this paragraph.

We have revised this paragraph of the manuscript to better clarify the logic. We agree that getting or obtaining a prior flux estimate is trivial; there are many options available for a prior flux estimate. However, the choice of among these models is not trivial. Furthermore, the prior flux estimate is not used directly in the equations for the posterior covariance matrix, but it does affect the posterior uncertainties through other means. If the prior flux model is skilled, the prior uncertainties should be be small (assuming a Bayesian inverse model), and the posterior uncertainties will be smaller than they would otherwise be. If the prior flux model is unskilled, the prior uncertainties should be large, and the posterior uncertainties will correspondingly be larger than they might otherwise be. Hence, the choice among prior models can have important implications. The posterior uncertainties may be small (or large) due to a skilled (or unskilled prior), and one could alias this effect for the capabilities of the observation network.

**p. 6, "Second, a modeler must estimate several individual sources of uncertainty as inputs to the inverse model." There is always statistical modeling involved and the present study is another example. Please remove this argument as well to only keep the last one (computational burden).**

We have clarified this text in the revised manuscript. Most top-down studies do use statistical models, but not all statistical models are the same. To build a Bayesian inverse model, one must differentiate between prior errors and errors in the atmospheric model and/or observations. Both sets of errors can have complex, non-stationary covariances. By contrast, we evaluate current OCO-2 observations using a statistical model that does not require differentiating among these types of errors. Furthermore, we estimate the covariances locally and therefore do not need to estimate the global, non-stationary

structure of the errors in one fell swoop.

**p. 9, l. 30: Liang et al. refer to a specific GOSAT L2 product. Other products would have different offset to GOSAT, or even no significant offset at all. Please remove.**

We have removed the reference as requested.

**p. 11, l. 18: please remove "4DVAR" from the sentence since the inversion technique is irrelevant to the argument.**

We have removed this term.

[revised manuscript text omitted]